# NAG-GS: Semi-Implicit, Accelerated and Robust Stochastic Optimizers

## Abstract

Classical machine learning models such as deep neural networks are usually trained by using Stochastic Gradient Descent-based (SGD) algorithms. The classical SGD can be interpreted as a discretization of the stochastic gradient flow. In this paper we propose a novel, robust and accelerated stochastic optimizer that relies on two key elements: (1) an accelerated Nesterov-like Stochastic Differential Equation (SDE) and (2) its semi-implicit Gauss-Seidel type discretization. The convergence and stability of the obtained method, referred to as NAG-GS, are first studied extensively in the case of the minimization of a quadratic function. This analysis allows us to come up with an optimal step size (or learning rate) in terms of rate of convergence while ensuring the stability of NAG-GS. This is achieved by the careful analysis of the spectral radius of the iteration matrix and the covariance matrix at stationarity with respect to all hyperparameters of our method. We show that NAG-GS is competitive with state-of-the-art methods such as momentum SGD with weight decay and AdamW for the training of machine learning models such as the logistic regression model, the residual networks models on standard computer vision datasets, and Transformers in the frame of the GLUE benchmark.

## 1 Introduction

Nowadays, machine learning, and more particularly deep learning, has achieved promising results on a wide spectrum of AI application domains. In order to process large amounts of data, most competitive approaches rely on the use of deep neural networks. Such models require to be trained and the process of training usually corresponds to solving a complex optimization problem. The development of fast methods is urgently needed to speed up the learning process and obtain efficiently trained models. In this paper, we introduce a new optimization framework for solving such problems.

**Main contributions of our paper:**

- We propose a new accelerated gradient method of Nesterov type for convex and non-convex stochastic optimization;
- We analyze the properties of the method both theoretically and experimentally;
- We show that our method is robust to the selection of hyperparameters, memory-efficient compared with AdamW and competitive with baseline methods in various benchmarks.

**Organization of our paper:**

- Section 1.1 gives the theoretical background for our method.
- In Section 2, we propose an accelerated system of Stochastic Differential Equations (SDE) and an appropriate solver that rely on a particular discretization of the SDE's system. The obtained method, referred to as NAG-GS (Nesterov Accelerated Gradient with Gauss-Seidel Splitting), is first discussed in terms of convergence in the simple but central case of quadratic functions. Moreover, we apply our method for solving a 1-dimensional non-convex SDE for which we bring strong numerical evidences of the superior acceleration allowed by NAG-GS method compared to classical SDE solvers, see Appendix B.
- In Section 3, NAG-GS is tested to tackle stochastic optimization problems of increasing complexity and dimension, starting from the logistic regression model to the training of large machine learning models such as ResNet20, ResNet50 and Transformers.

## 1.1 PRELIMINARIES

We start here with some general considerations in the deterministic setting for obtaining an accelerated Ordinary Differential Equations (ODE) that will be extended in the stochastic setting in Section 2.1. We consider iterative methods for solving the unconstrained minimization problem:

$$\min_{x \in V} f(x), \tag{1}$$

where $V$ is a Hilbert space, and $f : V \to \mathbb{R} \cup \{+\infty\}$ is a properly closed convex extended real-valued function. In the following, for simplicity, we shall consider the particular case of $\mathbb{R}^n$ for $V$ and consider function $f$ smooth on the entire space. We also suppose $V$ is equipped with the canonical inner product $\langle x, y \rangle = \sum_i^n x_i y_i$ and the correspondingly induced norm $\|x\| = \sqrt{\langle x, x \rangle}$. Finally, we will consider in this section the class of functions $\mathcal{S}_{L,\mu}^{1,1}$ which stands for the set of strongly convex functions of parameter $\mu > 0$ with Lipschitz-continuous gradients of constant $L > 0$. For such class of functions, it is well-known that the global minimizer exists uniquely Nesterov (2018). One well-known approach to derive the Gradient Descent (GD) method is discretizing the so-called gradient flow:

$$\dot{x}(t) = -\nabla f(x(t)), \quad t > 0. \tag{2}$$

The simplest forward (explicit) Euler method with step size $\alpha_k > 0$ leads to the GD method:

$$x_{k+1} \leftarrow x_k - \alpha_k \nabla f(x_k).$$

In the terminology of numerical analysis, it is well-known that this method is conditionally $A$-stable and for $f \in \mathcal{S}_{L,\mu}^{1,1}$ with $0 \leq \mu \leq L \leq \infty$, the step size $\alpha_k = 1/L$ is allowed to get linear rate of convergence. Note that the highest rate of convergence is achieved for $\alpha_k = \frac{2}{\mu+L}$. In this case:

$$\|x_k - x^\star\|^2 \leq \left(\frac{Q_f - 1}{Q_f + 1}\right)^{2k} \|x_0 - x^\star\|^2$$

with $Q_f = \frac{L}{\mu}$, usually referred to as the condition number of function $f$ Nesterov (2018). One can also consider the backward (implicit) Euler method:

$$x_{k+1} \leftarrow x_k - \alpha_k \nabla f(x_{k+1}), \tag{3}$$

which is unconditionally A-stable. Here-under, we summarize the methodology proposed by Luo & Chen (2021) to come up with a general family of accelerated gradient flows by focusing on the following simple problem:

$$\min_{x \in \mathbb{R}^n} f(x) = \frac{1}{2} x^T A x \tag{4}$$

for which the gradient flow in equation 2 reads simply as:

$$\dot{x}(t) = -Ax(t), \quad t > 0, \tag{5}$$

where $A$ is a $n$-by-$n$ symmetric positive semi-definite matrix ensuring that $f \in \mathcal{S}_{L,\mu}^{1,1}$ where $\mu$ and $L$ respectively correspond to the minimum and maximum eigenvalues of matrix $A$, which are real and positive by hypothesis. Instead of solving directly equation 5, authors of Luo & Chen (2021) turn to a general linear ODE system:

$$\dot{y}(t) = Gy(t), \quad t > 0. \tag{6}$$

The main idea consists in seeking such a system 6 with some asymmetric block matrix $G$ that transforms the spectrum of $A$ from the real line to the complex plane and reduces the condition number from $\kappa(A) = \frac{L}{\mu}$ to $\kappa(G) = O\left(\sqrt{\frac{L}{\mu}}\right)$. Afterwards, accelerated gradient methods can be constructed from A-stable methods for solving equation 6 with a significant larger step size and consequently improve the contraction rate from $O\left(\left(\frac{Q_f-1}{Q_f+1}\right)^{2k}\right)$ to $O\left(\left(\frac{\sqrt{Q_f}-1}{\sqrt{Q_f}+1}\right)^{2k}\right)$. Furthermore, to handle the convex case $\mu = 0$, authors from Luo & Chen (2021) combine the transformation idea with a suitable time scaling technique. In this paper we consider one transformation that relies on the embedding of $A$ into some $2 \times 2$ block matrix $G$ with a rotation built-in Luo & Chen (2021):

$$G_{NAG} = \begin{bmatrix} -I & I \\ \mu/\gamma - A/\gamma & -\mu/\gamma I \end{bmatrix} \tag{7}$$

where $\gamma$ is a positive time scaling factor which satisfies

$$\dot{\gamma} = \mu - \gamma, \quad \gamma(t=0) = \gamma_0 > 0. \tag{8}$$

Note that, given $A$ positive definite, we can easily show that for the considered transformation, we have that $\mathcal{R}(\lambda) < 0$ for all $\lambda \in \sigma(G)$ with $\sigma(G)$ denoting the spectrum of $G$, i.e. the set of all eigenvalues of $G$. Further, we will denote by $\rho(G) := \max_{\lambda \in \sigma(G)} |\lambda|$ the spectral radius of matrix $G$. Let us now consider the NAG block Matrix and let $y = (x, v)$, the dynamical system given in equation 6 with $y(0) = y_0 \in \mathbb{R}^{2n}$ reads:

$$\begin{aligned} \dot{x} &= v - x, \\ \dot{v} &= \frac{\mu}{\gamma}(x - v) - \frac{1}{\gamma}Ax \end{aligned} \tag{9}$$

with initial conditions $x(0) = x_0$ and $v(0) = v_0$. Before going further, let us remark that this linear ODE can be expressed as the following second-order ODE by eliminating $v$:

$$\gamma\ddot{x} + (\gamma + \mu)\dot{x} + Ax = 0, \tag{10}$$

where $Ax$ is therefore the gradient of $f$ w.r.t. $x$. Thus, one could generalize this approach for any function $f \in \mathcal{S}_{L,\mu}^{1,1}$ by replacing $A$ and $Ax$ by $\nabla f(x)$ respectively within equation 7, equation 9 and equation 10. Finally, some additional and useful insights are discussed in Appendix A.

## 2 MODEL AND THEORY

### 2.1 ACCELERATED STOCHASTIC GRADIENT FLOW

In the previous section, we presented a family of accelerated Gradient flows obtained by an appropriate spectral transformation $G$ of matrix $A$, see equation 9. One can observe the presence of a gradient term of the smooth function $f(x)$ at $x$ in the second differential equation. Let us recall that $Ax$ can be replaced by $\nabla f(x)$ for any function $f \in \mathcal{S}_{L,\mu}^{1,1}$. In the frame of this paper, function $f(x)$ may correspond to some loss function used to train neural networks. For such setting, we assume that the gradient input $\nabla f(x)$ is contaminated by noise due to finite-sample estimate of the gradient. The study of accelerated Gradient flows is now adapted to include and model the effect of the noise; to achieve this we consider the dynamics given in equation 6 perturbed by a general martingale process. This leads us to consider the following Accelerated Stochastic Gradient (ASG) flows:

$$\begin{aligned} \frac{dx}{dt} &= v - x, \\ \frac{dv}{dt} &= \frac{\mu}{\gamma}(x - v) - \frac{1}{\gamma}Ax + \frac{dZ}{dt}, \end{aligned} \tag{11}$$

which corresponds to an (Accelerated) system of SDE's, where $Z(t)$ is a continuous Ito martingale. We assume that $Z(t)$ has the simple expression $dZ = \sigma dW$, where $W = (W_1, ..., W_n)$ is a standard $n$-dimensional Brownian Motion. As a simple and first approach, we consider the volatility parameter $\sigma$ constant. In the next section, we present the discretizations considered for ASG flows given in equation 11.

### 2.2 DISCRETIZATION: GAUSS-SEIDEL SPLITTING AND SEMI-IMPLICITNESS

In this section, we present the main strategy to discretize the Accelerated SDE's system from equation 11. The main motivation behind the discretization method is to derive integration schemes that are, in the best case, unconditionally A-stable or conditionally A-stable with the highest possible integration step. In the classical terminology of (discrete) optimization methods, this amounts to ensure convergence of the obtained methods with the largest possible step size and consequently improve the contraction rate (or the rate of convergence). In Section 1.1, we have briefly recalled that the most well-known unconditionally A-stable scheme was the backward Euler method (see equation 3), which is an implicit method and hence can achieve faster convergence rate. However, this requires to either solve a linear system either, in the case of a general convex function, to compute the root of a non-linear equation, both situations leading to a high computational cost. This is the

main reason why few implicit schemes are used in practice for solving high-dimensional optimization problems. But still, it is expected that an explicit scheme closer to the implicit Euler method will have good stability with a larger step size than the one offered by a forward Euler method. Motivated by the Gauss–Seidel (GS) method for solving linear systems, we consider the matrix splitting $G = M + N$ with $M$ being the lower triangular part of $G$ and $N = G - M$, we propose the following Gauss-Seidel splitting scheme for equation 6 perturbated with noise:

$$\frac{y_{k+1} - y_k}{\alpha_k} = My_{k+1} + Ny_k + \begin{bmatrix} 0 \\ \sigma \frac{W_{k+1} - W_k}{\alpha_k} \end{bmatrix} \tag{12}$$

which for $G = G_{NAG}$ (see (7)), gives the following semi-implicit scheme with step size $\alpha_k > 0$:

$$\begin{aligned} \frac{x_{k+1} - x_k}{\alpha_k} &= v_k - x_{k+1}, \\ \frac{v_{k+1} - v_k}{\alpha_k} &= \frac{\mu}{\gamma_k}(x_{k+1} - v_{k+1}) - \frac{1}{\gamma_k}Ax_{k+1} + \sigma\frac{W_{k+1} - W_k}{\alpha_k}. \end{aligned} \tag{13}$$

Note that due to properties of Brownian motion we can simulate its values at the selected points by: $W_{k+1} = W_k + \Delta W_k$, where $\Delta W_k$ are independent random variables with distribution $\mathcal{N}(0, \alpha_k)$. On a practical point of view, we will use $\Delta W_k = \sqrt{\alpha_k}\eta_k$, where $\eta_k \sim \mathcal{N}(0, 1)$. Furthermore, ODE (8) corresponding to the parameter $\gamma$ is also discretized implicitly:

$$\frac{\gamma_{k+1} - \gamma_k}{\alpha_k} = \mu - \gamma_{k+1}, \quad \gamma_0 > 0. \tag{14}$$

As already mentioned earlier, heuristically, for general $f \in \mathcal{S}_{L,\mu}^{1,1}$ with $\mu \geq 0$, we just replace $Ax$ in equation 13 with $\nabla f(x)$ and obtain the following NAG-GS scheme:

$$\begin{aligned} \frac{x_{k+1} - x_k}{\alpha_k} &= v_k - x_{k+1}, \\ \frac{v_{k+1} - v_k}{\alpha_k} &= \frac{\mu}{\gamma_k}(x_{k+1} - v_{k+1}) - \frac{1}{\gamma_k}\nabla f(x_{k+1}) + \sigma\frac{W_{k+1} - W_k}{\alpha_k}. \end{aligned} \tag{15}$$

Finally, we come to the following methods, which are addressed as NAG-GS method (see Algorithm 1). There we assume that the gradient $\nabla f(x_{k+1})$ is computed with some unknown noise.

---

**Algorithm 1** Nesterov Accelerated Gradients with Gauss–Seidel splitting (NAG-GS).

---

**Input:** Choose point $x_0 \in \mathbb{R}^n$, some $\mu \geq 0, \gamma_0 > 0$.
  Set $v_0 := x_0$.
  **for** $k = 1, 2, \ldots$ **do**
    Choose step size $\alpha_k > 0$.
    ▷ Update parameters and state $x$:
    Set $a_k := \alpha_k(\alpha_k + 1)^{-1}$.
    Set $\gamma_{k+1} := (1 - a_k)\gamma_k + a_k\mu$.
    Set $x_{k+1} := (1 - a_k)x_k + a_k v_k$.
    ▷ Update state $v$:
    Set $b_k := \alpha_k\mu(\alpha_k\mu + \gamma_{k+1})^{-1}$.
    Set $v_{k+1} := (1 - b_k)v_k + b_k x_{k+1} - \mu^{-1}b_k\nabla f(x_{k+1})$.
  **end for**

---

Moreover, the step size update can be performed with different strategies, for instance one may choose the method proposed by Nesterov (Nesterov, 2018, Method 2.2.7) which specifies to compute $\alpha_k \in (0, 1)$ such that $L\alpha_k^2 = (1 - \alpha_k)\gamma_k + \alpha_k\mu$. Note that for $\gamma_0 = \mu$, hence the sequences $\gamma_k = \mu$ and $\alpha_k = \sqrt{\frac{\mu}{L}}$ for all $k \geq 0$. Other strategies are discussed in Luo & Chen (2021) such as $L\alpha_k^2 = \gamma_k(1 + \alpha_k)$. Finally, one could simply choose a constant step size variant of Algorithm 1, we select and discuss this approach in Section 2.3.

Let us mention that full-implicit discretizations have been considered and studied by the authors, these will be briefly discussed in Appendix A.2. However, their interests are, at the moment, limited for ML applications since the obtained implicit schemes are connected to a specific family of second-order methods which are intractable for real-life ML models.

Table 1: Summary on the comparison of NAG-GS to the reference optimizer for different neural architectures (greater is better). Target metrics are ACC@1 and ACC@5 for RESNET20 and RESNET50 respectively and average score on GLUE for ROBERTA.

| MODEL | DATASET | OPTIMIZER | SCORE |
|-------|---------|-----------|-------|
| RESNET20 | CIFAR-10 | SGD-MW | 91.25 |
| | | NAG-GS | **91.29** |
| RESNET50 | IMAGENET | SGD-MW | **92.70** |
| | | NAG-GS | 89.82 |
| ROBERTA | GLUE | ADAMW | **82.92** |
| | | NAG-GS | 81.30 |

### 2.3 CONVERGENCE ANALYSIS OF QUADRATIC CASE

We mentioned in previous section that one could choose a constant step size strategy for Algorithm 1. We propose to study how to select a maximum (constant) step size that ensures an optimal contraction rate while guaranteeing the convergence, or the stability of NAG-GS method once used to solve SDE's system 11. Ultimately, we show that the choice of the optimal (constant) step size is actually mostly influenced by the values of $\mu$, $L$ and $\gamma$. These (hyper)parameters are central and in order to show this, we study two key quantities, namely the spectral radius of the iteration matrix and the covariance matrix associated to NAG-GS method summarized by Algorithm 1. Note that this theoretical study only concerns the case $f(x) = \frac{1}{2}x^T A x$. Considering the size limitation of the paper, we present below only the main theoretical result and place its proof in Appendix A.1.4:

**Theorem 1** *For $G_{NAG}$ 7, given $\gamma \geq \mu$, and assuming $0 < \mu = \lambda_1 \leq \ldots \leq \lambda_n = L < \infty$; if $0 < \alpha \leq \frac{\mu+\gamma+\sqrt{(\mu-\gamma)^2+4\gamma L}}{L-\mu}$, then the NAG-GS method summarized by Algorithm 1 is convergent for the $n$-dimensional case, with $n > 2$.*

**Remark 1** *It is important to mention that theoretical result about the accelerated convergence for the stochastic setting **holds exactly in the same way as for the deterministic setting**.*

All the steps of the convergence analysis are fully detailed in Appendix A.1 and organized as follows:

- Appendices A.1.1 and A.1.2 respectively provide the full analysis of the spectral radius of the iteration matrix associated to NAG-GS method and the covariance matrix at stationarity w.r.t. all (hyper)parameters $\mu$, $L$, $\gamma$ and $\sigma$, for the dimensional case $n = 2$. The theoretical results obtained are summarized in Appendix A.1.3 to come up with an optimal (constant) step size in terms of contraction rate.
- Numerical tests are performed and detailed in Appendix A.1.5 to support the theoretical results obtained for the quadratic case.

## 3 EXPERIMENTS

We test NAG-GS method on several neural architectures: logistic regression, transformer models for natural language processing tasks, and residual networks for computer vision tasks. Section 3.1 presents the numerical results for logistic regression. Sections 3.2 and 3.3 focus on the tests carried out on transformers and residual networks. For these two neural architectures, in order to perform a benchmark of our method as fair as possible, we replace the reference optimizers with ours and adjust only hyperparameters of our optimizer. We keep intact the model architectures and model hyperparameters such as dropout rate, schedule, batch size, number of training epochs, evaluation methodology. Moreover, we carry out ablation study on small real-world models. The results of the benchmark for transformer models and residual networks are summarized in Table 1.

Furthermore, in Section 3.4, the experiments highlight the importance of updating $\gamma$ during the training process. In Section 3.5 we present some preliminary experiments to study the relations

between convergence and Hessian spectrum. Moreover, in Appendix C.2 we bring some numerical evidences that support the theoretical constraints on the optimizer parameters that have been derived in Section 2. The implementation details are discussed in Appendix C.3.

## 3.1 LOGISTIC REGRESSION

In this section, we benchmark NAG-GS method against state-of-the-art optimizers on the logistic regression training problem for MNIST dataset LeCun et al. (2010). Since this problem is convex and non-quadratic, we consider this problem as the natural and next test case after the theoretical analysis and numerical tests of NAG-GS method in Section 2.3 for the quadratic convex problem. In Figure 2 we present the comparison of NAG-GS method with competitors. We confirm numerically that NAG-GS method allows the use of a larger range of values for the learning rate than SGD Momentum and AdamW optimizers, highlighting the robustness of our method w.r.t. the selection of hyperparameters. Moreover, the results indicate that the semi-implicit nature of NAG-GS method indeed ensures the acceleration effect through the use of larger learning rates while keeping a good accuracy of the model, and this holds not only for the convex quadratic problems but also for non-quadratic convex ones.

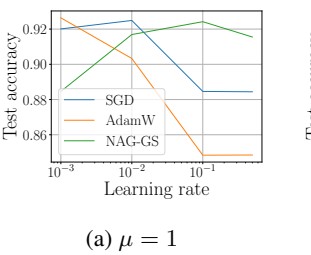

(a) $\mu = 1$

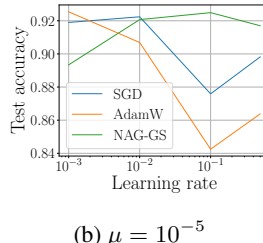

(b) $\mu = 10^{-5}$

Figure 2: Dependence of the test accuracy on the learning rates for the considered methods. NAG-GS provides the highest test accuracy for larger step size.

Table 2: Test accuracies for NAG-GS, SGD-Momentum and AdamW.

| lr | NAG-GS | SGD | AdamW |
|---|---|---|---|
| $10^{-3}$ | 0.8934 | 0.9190 | **0.9254** |
| $10^{-2}$ | 0.9207 | **0.9224** | 0.9069 |
| 0.1 | **0.9249** | 0.8759 | 0.8425 |
| 0.5 | **0.9170** | 0.8982 | 0.8638 |

## 3.2 TRANSFORMER MODELS

In this section we test NAG-GS optimizer in the frame of natural language processing for the tasks of fine-tuning pretrained model on GLUE benchmark datasets Wang et al. (2018). We use pretrained RoBERTa Liu et al. (2019) model from Hugging Face's TRANSFORMERS Wolf et al. (2020) library. In this benchmark, the reference optimizer is AdamW Ilya et al. (2019) with polynomial learning rate schedule. The training setup defined in Liu et al. (2019) is used for both NAG-GS and AdamW optimizers. We search for an optimal learning rate for NAG-GS optimizer with fixed $\gamma$ and $\mu$ to get the best performance on the task at hand. Note that NAG-GS is used with constant schedule which makes it simpler to tune. In terms of learning rate values, the one allowed by AdamW is around $10^{-5}$ while NAG-GS allows a much bigger value of $10^{-2}$. Evaluation results on GLUE tasks are presented in Table 3. Despite a rather restraint search space for NAG-GS hyperparameters, it demonstrates better performance on some tasks and worse performance on others. Figure 3 shows the behavior of loss values and target metrics on GLUE (see Appendix C).

Table 3: Comparison of AdamW and NAG-GS optimizers in fine-tuning on GLUE benchmark. We use reported hyperparameters for AdamW. In case of NAG-GS we search hyperparameters space for the best performance metric. Search space consists of learning rate $\alpha$ from $[10^{-3}, 10^0]$, factor $\gamma$ from $[10^{-2}, 10^0]$, and momentum $\mu = 1$.

| OPTIMIZER | CoLA | MNLI | MRPC | QNLI | QQP | RTE | SST2 | STS-B | WNLI |
|---|---|---|---|---|---|---|---|---|---|
| ADAMW | **61.60** | **87.56** | 88.24 | **92.62** | **91.69** | **78.34** | **94.95** | **90.68** | **56.34** |
| NAG-GS | **61.60** | 87.24 | **90.69** | 92.42 | 91.01 | 73.65 | 94.50 | 90.21 | **56.34** |

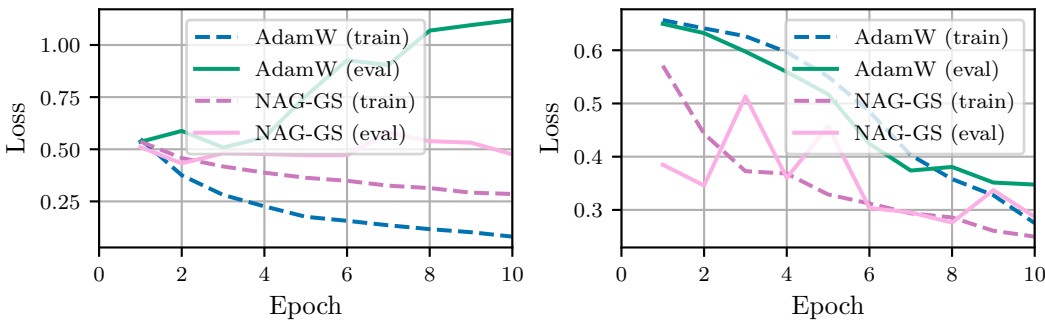

Figure 3: Cross-entropy losses on validation and train sets for COLA (left) and MRPC (right) tasks. Solid lines corresponds to the best trial with NAG-GS optimizer.

### 3.3 RESIDUAL NETWORKS

We compare NAG-GS to momentum SGD with weight decay (SGD-MW) on residual networks (ResNets) He et al. (2016). We choose ResNet20 for versatile experimental verification of properties of our optimizer and ResNet50 to test scaling of NAG-GS on large datasets and models which are used on practice.

**ResNet50.** Due to limitation of computational resources, we train ResNet50 on ImageNet Deng et al. (2009) for a few learning rate values along with fixed $\gamma$ and $\mu$. We select the best overall performing parameters for our optimizer while, for the reference optimizer SGD-MW, we use the parameters reported in the literature. Also, we found that, unlike fine-tuning on Transformers, piecewise-constant schedule gives better target metrics than constant one. The best acc@5 performance for NAG-GS is within three percentage points worse than for SGD-MW (see Table 1, and Figure 4b).

**ResNet20.** Since ResNet20 has lesser number of parameters than ResNet50, we carried out more intensive experiments in order to evaluate more deeply the performance of NAG-GS for computer vision tasks (residual networks in particular) and to show that NAG-GS with appropriate choice of optimizer parameters is on par with SGD-MW (see Table 1, and Figure 4a). The classification problem is solved using CIFAR-10 Krizhevsky (2009). Experimental setup is the same in all experiments except optimizer and its parameters. The best test score for NAG-GS is achieved for $\alpha = 0.11$, $\gamma = 17$, and $\mu = 0.01$.

### 3.4 UPDATABLE SCALING FACTOR $\gamma$

According to the theory of NAG-GS optimizer presented in Section 2, the scaling factor $\gamma$ decays exponentially fast to $\mu$ and, in the case $\gamma_0 = \mu$, $\gamma$ remains constant along iterations. So, a natural question arises: is the update on $\gamma$ necessary? Our experiments confirm that scaling factor $\gamma$ should be updated accordingly to Algorithm 1, even in this highly non-convex setting, in order to get better metrics on test sets.

We use experimental setup for ResNet20 from Section 3.3 and search for hyperparameters for NAG-GS with updatable $\gamma$ and with constant one. Common hyperoptimization library OPTUNA Akiba et al. (2019) are used with a budget of 160 iterations to sample NAG-GS parameters. Figure 5 plots the evolution of the best score value along optimization time. The final difference is about 0.5 which is a significant difference in terms of final classification error.

### 3.5 NON-CONVEXITY AND HESSIAN SPECTRUM

Theoretical analysis of NAG-GS highlights the importance of the smallest eigenvalue of the Hessian matrix for convex and strongly convex functions. Unfortunately, the objective functions usually considered for the training of neural networks are not convex. In this section we try to address this issue. The smallest model in our experimental setup is ResNet20. However, we cannot afford to compute exactly the Hessian matrix since ResNet20 has almost 300k parameters. Instead, we use Hessian-vector product (HVP) $H(x)$ and apply matrix-free algorithms for finding the extreme eigenvalues. We estimate the extreme eigenvalues of the Hessian spectrum with power iterations

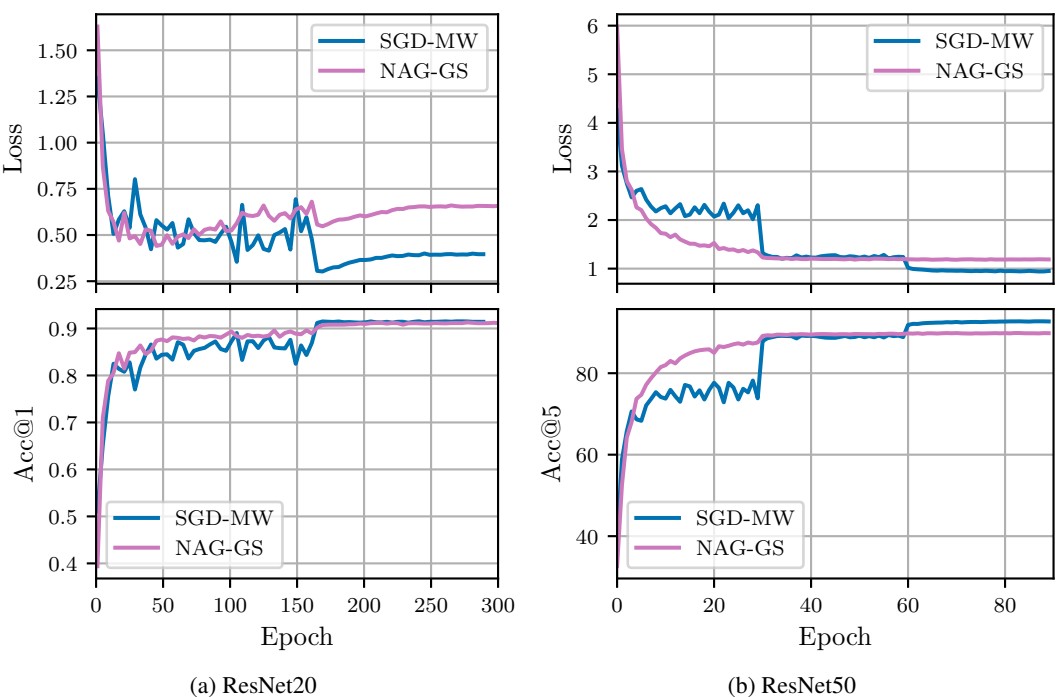

(a) ResNet20             (b) ResNet50

Figure 4: Evaluation of NAG-GS with SGD-MW on ResNet-20 (left) and ResNet-50 (right) on CIFAR-10 and ImageNet respectively.

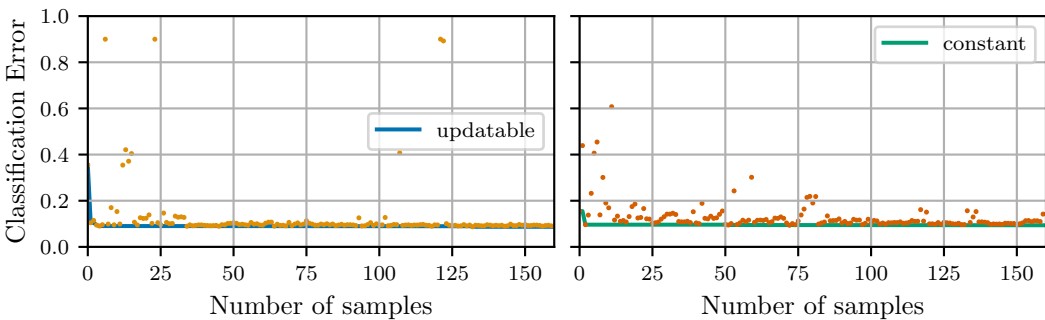

Figure 5: The best acc@1 on test set for updatable and fixed scaling factor $\gamma$ during hyperoptimization. NAG-GS with updatable $\gamma$ gives more frequently better results than the ones obtained with constant $\gamma$. The final difference is about $0.5$.

(PI) along with Rayleight quotient (RQ) Golub & van Loan (2013). PI is used to get a good initial vector which is used later in the optimization of RQ. In order to get more useful initial vector for the estimation of the smallest eigenvalue, we apply the spectral shift $H(x) - \lambda_{\max} x$ and use the corresponding eigenvector.

Figure 6 shows the extreme eigenvalues of ResNet20 Hessian at the end of each epoch for the batch size 256 in the same setup as in Section 3.3. The largest eigenvalue is strictly positive while the smallest one is negative and usually oscillates around $\mu = -1$. It turns out that there is an island of hyperparameters in the vicinity of that $\mu$. We report that training ResNet20 with hyperparameters included in this island gives good target metrics. The domain of negative momenta is non-conventional and not well understood, to the best of our knowledge. Moreover, there is no theoretical guarantees for NAG-GS in the non-convex case and negative $\mu$. However, Velikanov et al. (2022) reports the existence of regions of convergence for SGD with negative momentum, which supports our observations. The theoretical aspects of these observations will be studied in further works.

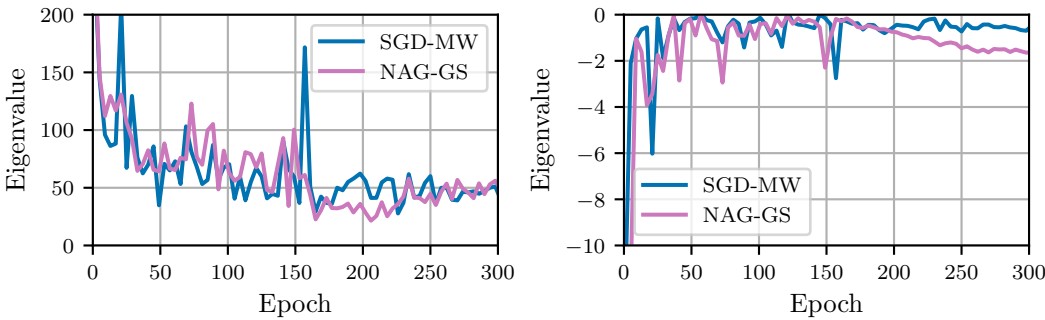

Figure 6: Evolution of the extreme eigenvalues (the largest and the smallest ones) during training RESNET20 on CIFAR-10 with NAG-GS.

## 4 RELATED WORKS

The approach of interpreting and analyzing optimization methods from the ODEs discretization perspective is well-known and widely used in practice Muehlebach & Jordan (2019); Wilson et al. (2021); Shi et al. (2021). The main advantage of this approach is to construct a direct correspondence between the properties of some classes of ODEs and their associated optimization methods. In particular, gradient descent and Nesterov accelerated methods are discussed in Su et al. (2014) as a particular discretization of ODEs. In the same perspective, many other optimization methods were analyzed, we can mention the mirror descent method and its accelerated versions Krichene et al. (2015), the proximal methods Attouch et al. (2019) and ADMM Franca et al. (2018). It is well known that discretization strategy is essential for transforming a particular ODE to an efficient optimization method, Shi et al. (2019); Zhang et al. (2018) investigate the most proper discretization techniques for different classes of ODEs. A similar analysis but for stochastic first-order methods is presented in Laborde & Oberman (2020); Malladi et al. (2022).

## 5 CONCLUSIONS AND FURTHER WORKS

We have presented a new and theoretically motivated stochastic optimizer called NAG-GS. It comes from the semi-implicit Gauss-Seidel type discretization of a well-chosen accelerated Nesterov-like SDE. These building blocks ensure two central properties for NAG-GS: (1) the ability to accelerate the optimization process and (2) a great robustness to the selection of hyperparameters, in particular for the choice of the learning rate. We demonstrate these features theoretically and provide detailed analysis of the convergence of the method in the quadratic case. Moreover, we show that NAG-GS is competitive with state-of-the-art methods for tackling a wide variety of stochastic optimization problems of increasing complexity and dimension, starting from the logistic regression model to the training of large machine learning models such as ResNet20, ResNet50 and Transformers. In all tests, NAG-GS demonstrates better or competitive performance compared with standard optimizers. Further works will focus on the theoretical analysis of NAG-GS in the non-convex case and the derivation of efficient and tractable higher-order methods based on the full-implicit discretization of the accelerated Nesterov-like SDE.

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

## A  ADDITIONAL REMARKS RELATED TO THEORETICAL BACKGROUND

An accelerated ODE has been presented in Section 1.1 which relied on a specific spectral transformation. In this brief Appendix, we add some useful insights:

- Equation 10 is a variant of heavy ball model with variable damping coefficients in front of $\ddot{x}$ and $\dot{x}$.

- Thanks to the scaling factor $\gamma$, both the convex case $\mu = 0$ and the strongly convex case $\mu > 0$ can be handled in a unified way.

- In the continuous time, one can solve easily equation 8 as follows: $\gamma(t) = \mu + (\gamma_0 - \mu)e^{-t}$, $t \geq 0$. Since $\gamma_0 > 0$, we have that $\gamma(t) > 0$ for all $t \geq 0$ and $\gamma(t)$ converges to $\mu$ exponentially and monotonically as $t \to +\infty$. In particular, if $\gamma_0 = \mu > 0$, then $\gamma(t) = \mu$ for all $t \geq 0$. We remark here the links between the behavior of the scaling factor $\gamma(t)$ and the sequence $\{\gamma_k\}_{k=0}^{\infty}$ introduced by Nesterov Nesterov (2018) in its analysis of optimal first-order methods in discrete-time, see (Nesterov, 2018, Lemma 2.2.3).

- Authors from Luo & Chen (2021) prove the exponential decay property $\mathcal{L}(t) \leq e^{-t}\mathcal{L}_0$, $t > 0$ for a Taylored Lyapunov function $\mathcal{L}(t) := f(x(t)) - f(x^\star) + \frac{\gamma(t)}{2}\|v(t) - x^\star\|^2$ where $x^\star \in \operatorname{argmin} f$ is a global minimizer of $f$. Again we note the similarity between the Lyapunov function proposed here and the estimating sequence $\{\phi_k(x)\}_{k=0}^{\infty}$ of function $f$ introduced by Nesterov in its optimal first-order methods analysis Nesterov (2018). In (Nesterov, 2018, Lemma 2.2.3), this sequence that takes the form $\phi_k(x) = \phi_k^\star(x) + \frac{\gamma_k}{2}\|v_k - x\|^2$ where $\gamma_{k+1} := (1 - \alpha_k)\gamma_k + \alpha_k\mu$ and $v_{k+1} := \frac{1}{\gamma_{k+1}}[(1 - \alpha_k)\gamma_k v_k + \alpha_k\mu y_k - \alpha_k\nabla f(y_k)]$ which stand for a forward Euler discretization respectively of equation 8 and second ODE of equation 9.

We ask the attentive reader to remember that this discussion mainly concerns the continuous time case. A second central part of our analysis was based on the methods of discretization of equation 9. Indeed, these discretization's ensure together with the spectral transformation 7 the optimal convergence rates of the methods and their particular ability to handle noisy gradients.

### A.1  CONVERGENCE/STABILITY ANALYSIS OF QUADRATIC CASE: DETAILS

As briefly mentioned in Section 2.3, the two key elements to come up with a maximum (constant) step size for Algorithm 1 are the study of the spectral radius of iteration matrix associated to NAG-GS scheme (Appendix A.1.1) and the covariance matrix at stationarity (Appendix A.1.2) w.r.t. all the significant parameters of the scheme that are : the step size (integration step/time step) $\alpha$, the convexity parameters $0 \leq \mu \leq L \leq \infty$ of function $f(x)$, the volatility of the noise $\sigma$ and the positive scaling parameter $\gamma$. Note that this theoretical study only concerns the case $f(x) = \frac{1}{2}x^T Ax$. Among others results, we will show in particular that for specific intervals for $\mu$ and $L$, a higher step size for $\alpha$ can be reached while ensuring the $A$-stability of NAG-GS method. Let remark finally that the two following sections consider the special case $n = 2$, the theoretical results obtained are summarized in Appendix A.1.3 and finally generalized for $n$-dimensional case in Appendix A.1.4 with a global convergence results for NAG-GS method.

### A.1.1  SPECTRAL RADIUS ANALYSIS

Let us assume $f(x) = \frac{1}{2}x^\top Ax$ and since $A \in \mathbb{S}_+^n$ by hypothesis, it is diagonalizable and can be presented as $A = \operatorname{diag}(\lambda_1, \ldots, \lambda_n)$ without loss of generality, that is to say that we will consider a system of coordinates composed of the eigenvectors of matrix $A$. Let us note that $\mu = \lambda_1 \leq \cdots \leq \lambda_i \leq \cdots \leq \lambda_n = L$.

In this setting, $y = (x, v) \in \mathbb{R}^4$ and the matrices $M$ and $N$ from the Gauss-Seidel Splitting of $G_{NAG}$ equation 7 are:

$$M = \begin{bmatrix} -I_{2\times2} & 0_{2\times2} \\ \mu/\gamma I_{2\times2} - A/\gamma & -\mu/\gamma I_{2\times2} \end{bmatrix} = \begin{bmatrix} -1 & 0 & 0 & 0 \\ 0 & -1 & 0 & 0 \\ 0 & 0 & -\mu/\gamma & 0 \\ 0 & \mu/\gamma - L/\gamma & 0 & -\mu/\gamma \end{bmatrix},$$

$$N = \begin{bmatrix} 0_{2\times2} & I_{2\times2} \\ 0_{2\times2} & 0_{2\times2} \end{bmatrix}$$

For the minimization of $f(x) = \frac{1}{2}x^\top A x$, given the property of Brownian motion $\Delta W_k = W_{k+1} - W_k = \sqrt{\alpha_k}\eta_k$ where $\eta_k \sim \mathcal{N}(0,1)$, equation 12 reads:

$$y_{k+1} = (I_{4\times4} - \alpha M)^{-1}(I_{4\times4} + \alpha N)y_k + (I_{4\times4} - \alpha M)^{-1}\begin{bmatrix} 0 \\ \sigma\sqrt{\alpha}\eta_k \end{bmatrix} \tag{16}$$

Since matrix $M$ is lower-triangular, matrix $I_{4\times4} - \alpha M$ is as well and can be factorized as follows:

$$I_{4\times4} - \alpha M = DT$$

$$= \begin{bmatrix} (1+\alpha)I_{2\times2} & 0_{2\times2} \\ 0_{2\times2} & (1+\frac{\alpha\mu}{\gamma})I_{2\times2} \end{bmatrix}\begin{bmatrix} I_{2\times2} & 0_{2\times2} \\ \frac{\alpha(A-\mu I_{2\times2})}{\gamma(1+\frac{\alpha\mu}{\gamma})} & I_{2\times2} \end{bmatrix}$$

Hence $(I_{4\times4} - \alpha M)^{-1} = T^{-1}D^{-1}$ where $D^{-1}$ can be easily computed. It remains to compute $T^{-1}$; $T$ can be decomposed as follows: $T = I_{4\times4} + Q$ with $Q$ a nilpotent matrix such that $QQ = O_{4\times4}$. For such decomposition, it is well known that:

$$T^{-1} = (I_{4\times4} + Q)^{-1} = I_{4\times4} - Q = \begin{bmatrix} I_{2\times2} & 0_{2\times2} \\ \frac{\alpha(\mu I_{2\times2} - A)}{\gamma(1+\tau_k)} & I_{2\times2} \end{bmatrix} \tag{17}$$

where $\tau_k = \frac{\alpha\mu}{\gamma}$. Combining these results, equation 16 finally reads:

$$y_{k+1} = \begin{bmatrix} \frac{1}{\alpha+1} & 0 & \frac{\alpha}{1+\alpha} & 0 \\ 0 & \frac{1}{\alpha+1} & 0 & \frac{\alpha}{1+\alpha} \\ 0 & 0 & \frac{1}{1+\tau} & 0 \\ 0 & \frac{\alpha(\mu-L)}{\gamma(\tau+1)(\alpha+1)} & 0 & \frac{\alpha^2(\mu-L)}{\gamma(1+\tau)(1+\alpha)} + \frac{1}{1+\tau} \end{bmatrix} y_k + \begin{bmatrix} 0 \\ \sigma\frac{\sqrt{\alpha}}{1+\tau}\eta_k \end{bmatrix} \tag{18}$$

$$= Ey_k + \begin{bmatrix} 0 \\ \sigma\frac{\sqrt{\alpha}}{1+\tau}\eta_k \end{bmatrix}$$

with $E$ denoting the iteration matrix associated to the NAG-GS method. Hence equation 18 includes two terms, the first is the product of the iteration matrix times the current vector $y_k$ and the second one features the effect of the noise. For the latter, it will be studied in Appendix A.1.2 on the point of view of maximum step size for the NAG-GS method through the key quantity of covariance matrix. Let us focus on the first term; it is clear that in order to get the maximum contraction rate, we should look for $\alpha$ that minimize the spectrum radius of $E$. Since the spectral radius is the maximum of absolute value of the eigenvalues of iteration matrix $E$, we start by computing them. Let us find the expression of $\lambda_i \in \sigma(E)$ for $1 \le i \le 4$ that satisfy $\det(E - \lambda I_{4\times4}) = 0$ as functions of the scheme's parameters. Solving

$$\det(E - \lambda I_{4\times4}) = 0$$
$$\equiv \frac{(\gamma\lambda - \gamma + \alpha\lambda\mu)(\lambda + \alpha\lambda - 1)(\gamma - 2\gamma\lambda + \gamma\lambda^2 + \alpha^2\lambda^2\mu - \alpha\gamma\lambda - \alpha\lambda\mu + L\alpha^2\lambda + \alpha\gamma\lambda^2 + \alpha\lambda^2\mu - \alpha^2\lambda\mu)}{(\alpha+1)^2(\gamma+\alpha\mu)^2} = 0 \tag{19}$$

leads to the following eigenvalues:

$$\lambda_1 = \frac{\gamma}{\gamma + \alpha\mu}$$

$$\lambda_2 = \frac{1}{1+\alpha}$$

$$\lambda_3 = \frac{2\gamma + \alpha\gamma + \alpha\mu - L\alpha^2 + \alpha^2\mu}{2(\gamma + \alpha\gamma + \alpha\mu + \alpha^2\mu)} +$$
$$\frac{\alpha\sqrt{L^2\alpha^2 - 2L\alpha^2\mu - 2L\alpha\mu - 2\gamma L\alpha - 4\gamma L + \alpha^2\mu^2 + 2\alpha\mu^2 + 2\gamma\alpha\mu + \mu^2 + 2\gamma\mu + \gamma^2}}{2(\gamma + \alpha\gamma + \alpha\mu + \alpha^2\mu)}$$

$$\lambda_4 = \frac{2\gamma + \alpha\gamma + \alpha\mu - L\alpha^2 + \alpha^2\mu}{2(\gamma + \alpha\gamma + \alpha\mu + \alpha^2\mu)} -$$
$$\frac{\alpha\sqrt{L^2\alpha^2 - 2L\alpha^2\mu - 2L\alpha\mu - 2\gamma L\alpha - 4\gamma L + \alpha^2\mu^2 + 2\alpha\mu^2 + 2\gamma\alpha\mu + \mu^2 + 2\gamma\mu + \gamma^2}}{2(\gamma + \alpha\gamma + \alpha\mu + \alpha^2\mu)}$$

$$\tag{20}$$

Let us first mention some general behavior or these eigenvalues: Given $\gamma$ and $\mu$ positive, we observe that:

1. $\lambda_1$ and $\lambda_2$ are positive decreasing functions w.r.t. $\alpha$. Moreover, for bounded $\gamma$ and $\mu$, we have $\lim_{\alpha \to \infty} |\lambda_1(\alpha)| = 0 = \lim_{\alpha \to \infty} |\lambda_2(\alpha)|$.

2. one can show that for $\alpha \in \left[\frac{\mu+\gamma-2\sqrt{\gamma L}}{L-\mu}, \frac{\mu+\gamma+2\sqrt{\gamma L}}{L-\mu}\right]$, functions $\lambda_3(\alpha)$ and $\lambda_4(\alpha)$ are complex values and one can easily show that both share the same absolute value. Note that the lower bound of the interval $\frac{\mu+\gamma-2\sqrt{\gamma L}}{L-\mu}$ is negative as soon as $\gamma \in [2L - \mu - 2\sqrt{L^2 - \mu L}, 2L - \mu + 2\sqrt{L^2 - \mu L}] \subseteq \mathbb{R}_+$. Moreover, one can easily show that $\lim_{\alpha \to \infty} |\lambda_3(\alpha)| = 0$ and $\lim_{\alpha \to \infty} |\lambda_4(\alpha)| = \frac{L-\mu}{\mu} = \kappa(A) - 1$. The latter limit shows that eigenvalues $\lambda_4$ plays a central role in the convergence of the NAG-GS method since it is the one that can reach the value one and violate the convergence condition, as soon as $\kappa(A) > 2$. The analysis of $\lambda_4$ also allows us to come up with a good candidate for the step size $\alpha$ that minimize the spectral radius of matrix $E$, especially and obviously at critical point $\alpha_{max} = \frac{\mu+\gamma+2\sqrt{\gamma L}}{L-\mu}$ which is positive since $L \geq \mu$ by hypothesis. Note that the case $L \to \mu$ gives some preliminary hints that the maximum step size can be almost "unbounded" in some particular case.

Now, let us study these eigenvalues in more details, it seems that three different scenarios must be studied:

1. For any variant of Algorithm 1 for which $\gamma_0 = \mu$, then $\gamma = \mu$ for all $k \geq 0$ and therefore $\lambda_1(\alpha) = \lambda_2(\alpha)$. Moreover, at $\alpha = \frac{\mu+\gamma+2\sqrt{\gamma L}}{L-\mu} = \frac{2\mu+2\sqrt{\mu L}}{L-\mu}$, we can easily check that $|\lambda_1(\alpha)| = |\lambda_2(\alpha)| = |\lambda_3(\alpha)| = |\lambda_4(\alpha)|$. Therefore $\alpha = \frac{2\mu+2\sqrt{\mu L}}{L-\mu}$ is the step size ensuring the minimal spectral radius and hence the maximum contraction rate. Figure 7 shows the evolution of the absolute values of the eigenvalues of iteration matrix $E$ w.r.t. $\alpha$ for such setting.

2. As soon as $\gamma < \mu$, one can easily show that $\lambda_1(\alpha) < \lambda_2(\alpha)$. Therefore the step size $\alpha$ with the minimal spectral radius is such that $|\lambda_4(\alpha)| = |\lambda_2(\alpha)|$. One can show that the equality holds for $\alpha = \frac{\mu+\gamma+\sqrt{(\mu-\gamma)^2+4\gamma L}}{L-\mu}$. One can easily check that $\frac{\mu+\gamma+\sqrt{(\mu-\gamma)^2+4\gamma L}}{L-\mu} - \frac{\mu+\gamma+2\sqrt{\gamma L}}{L-\mu} = (\mu-\gamma)^2 > 0$. Hence the second candidate for step size $\alpha$ will be bigger than the first one and the distance between them increasing as the squared distance between $\gamma$ and $\mu$. Figure 8 shows the evolution of the absolute values of the eigenvalues of iteration matrix $E$ w.r.t. $\alpha$ for this setting.

3. For $\gamma > \mu$: the analysis of this setting gives the same results as the previous point. According to Algorithm 1, $\gamma$ is either constant and equal to $\mu$ either decreasing to $\mu$ along iterations. Hence, the case $\gamma > \mu$ will be considered for the theoretical analysis when $\gamma \neq \mu$.

As a first summary, the detailed analysis of the eigenvalues of iteration matrix $E$ w.r.t. the significant parameters of the NAG-GS method leads us to come up with two candidates for the step size that minimize the spectral radius of $E$, hence ensuring the highest contraction rate possible. These results will be gathered with those obtained in Appendix A.1.2 dedicated to the covariance matrix analysis.

Let us now look at the behavior of the dynamics in expectation; given the properties of the Brownian motion and by applying the Expectation operator $\mathbb{E}$ on both sides of the system of SDE's 11, the resulting "averaged" equations identify with the "deterministic" setting studied by Luo & Chen (2021). For such setting, authors from Luo & Chen (2021) demonstrated that, if $0 \leq \alpha \leq \frac{2}{\sqrt{\kappa(A)}}$, then a Gauss–Seidel splitting-based scheme for solving equation 9 is A-stable for quadratic objectives in the deterministic setting. We conclude this section by showing that our two candidates we derived above for step size are higher than the limit $\frac{2}{\sqrt{\kappa(A)}}$ given in (Luo & Chen, 2021, Theorem 1). It can be intuitively understood in the case $L \to \mu$, however we give a formal proof in Lemma 1.

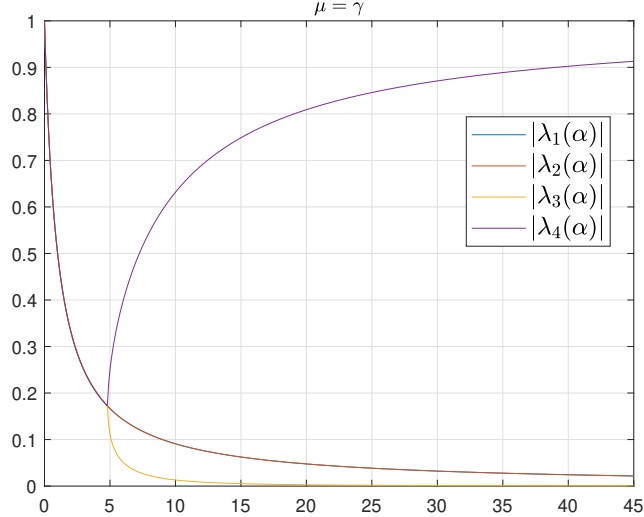

Figure 7: Evolution of absolute values of $\lambda_i$ w.r.t $\alpha$; $\mu = \gamma$.

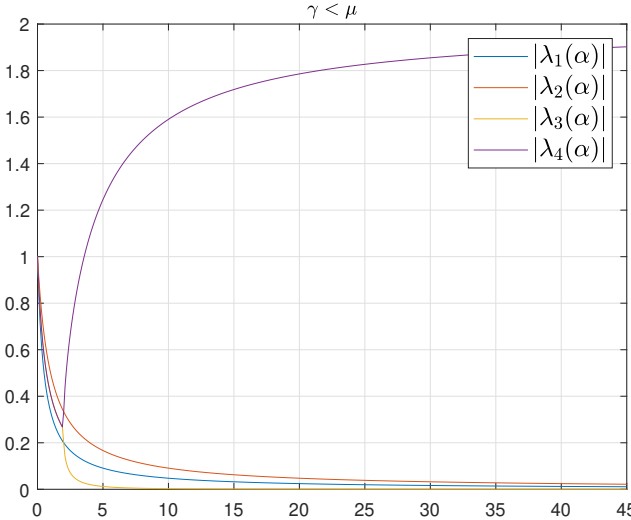

Figure 8: Evolution of absolute values of $\lambda_i$ w.r.t $\alpha$; $\gamma < \mu$.

**Lemma 1** *Given $\gamma > 0$, and assuming $0 < \mu < L$, then for $\gamma = \mu$ and $\gamma > \mu$ the following inequalities respectively hold:*

$$
\begin{aligned}
\frac{2\mu + 2\sqrt{\mu L}}{L - \mu} &> \frac{2}{\sqrt{\kappa(A)}} \\
\frac{\mu + \gamma + \sqrt{(\mu - \gamma)^2 + 4\gamma L}}{L - \mu} &> \frac{2}{\sqrt{\kappa(A)}}
\end{aligned}
\tag{21}
$$

*where $\kappa(A) = \frac{L}{\mu}$.*

**Proof:** Let us start for the case $\mu = \gamma$, hence first inequality from equation 21 becomes:

$$\frac{2\mu + 2\sqrt{L\mu}}{L - \mu} > \frac{2}{\sqrt{L/\mu}}$$
$$\equiv (\mu + \sqrt{L\mu})\sqrt{L/\mu} > (L - \mu)$$
$$\equiv \sqrt{\mu L} + L > L - \mu$$
$$\equiv \sqrt{\mu L} > -\mu$$

which holds for any positive $\mu, L$ and satisfied by hypothesis. For the case $\gamma > \mu$, we have:

$$\frac{\mu + \gamma + \sqrt{(\mu - \gamma)^2 + 4\gamma L}}{L - \mu} > \frac{2}{\sqrt{L/\mu}}$$

$$\equiv \sqrt{(\mu - \gamma)^2 + 4\gamma L} > \frac{2}{\sqrt{L/\mu}}(L - \mu) - \gamma - \mu$$

$$\equiv (\mu - \gamma)^2 + 4\gamma L > (\mu + 2\sqrt{\frac{\mu}{L}}(\mu - L) + \gamma)^2$$

$$\equiv \gamma > \frac{-2\mu^2 + \mu^3/L + \mu^2\sqrt{\mu/L} + \mu L - \mu L\sqrt{\mu/L}}{-\mu - \sqrt{\mu/L}(\mu - L) + L}$$

where second inequality hold since $L \geq \mu$ and last inequality holds since $-\mu - \sqrt{\mu/L}(\mu - L) + L > 0$ (one can easily check this by using $L > \mu$). It remains to show that:

$$\mu > \frac{-2\mu^2 + \mu^3/L + \mu^2\sqrt{\mu/L} + \mu L - \mu L\sqrt{\mu/L}}{-\mu - \sqrt{\mu/L}(\mu - L) + L}$$

which holds for any $\mu$ and $L$ positive (technical details are skipped; it mainly consists in the study of a table of signs of a polynomial equation in $\mu$).

Since $\gamma > \mu$ by hypothesis, therefore inequality

$$\gamma > \frac{-2\mu^2 + \mu^3/L + \mu^2\sqrt{\mu/L} + \mu L - \mu L\sqrt{\mu/L}}{-\mu - \sqrt{\mu/L}(\mu - L) + L}$$

holds for any $\mu$ and $L$ positive as well, conditions satisfied by hypothesis. This concludes the proof.

$\square$

Furthermore, let us note that both step size candidates, that are $\{\frac{2\mu + 2\sqrt{\mu L}}{L - \mu}, \frac{\mu + \gamma + \sqrt{(\mu - \gamma)^2 + 4\gamma L}}{L - \mu}\}$ respectively for the cases $\gamma = \mu$ and $\gamma > \mu$ show that NAG-GS method converges in the case $L \to \mu$ with a step size that tends to $\infty$, this behavior cannot be anticipated by the upper-bound given by (Luo & Chen, 2021, Theorem 1). Some simple numerical experiments are performed in Appendix A.1.5 to support this theoretical result.

Finally, based on previous discussions, let remark that for $\alpha \in [\frac{\mu + \gamma + \sqrt{(\mu - \gamma)^2 + 4\gamma L}}{L - \mu}, \infty]$ when $\gamma \neq \mu$ or $\alpha \in [\frac{2\mu + 2\sqrt{\mu L}}{L - \mu}, \infty]$ when $\gamma = \mu$, we have $\rho(E(\alpha)) = |\lambda_4(\alpha)|$ and one can show that $\rho(E)$ is strictly monotonically increasing function of $\alpha$ for all $L > \mu > 0$ and $\gamma > 0$ (see Appendix A.1.6 for the discussion).

### A.1.2 COVARIANCE ANALYSIS

In this section, we study the contribution to the computation of maximum step size for NAG-GS method through the analysis of covariance matrix at stationarity. Let us start by computing the covariance matrix $C$ obtained at iteration $k + 1$ from Algorithm 1:

$$C_{k+1} = \mathbb{E}(y_{k+1}y_{k+1}^T) \tag{22}$$

By denoting $\xi_k = \begin{bmatrix} 0 \\ \sigma\frac{\sqrt{\alpha}}{1+\tau}\eta_k \end{bmatrix}$, let us replace $y_{k+1}$ by its expression given in equation 18, equation 22 writes:

$$\begin{aligned} C_{k+1} &= \mathbb{E}(y_{k+1}y_{k+1}^T) \\ &= \mathbb{E}\left((Ey_k + \xi_k)(Ey_k + \xi_k)^T\right) \\ &= \mathbb{E}\left(Ey_ky_k^T E^T\right) + \mathbb{E}\left(\xi_k\xi_k^T\right) \end{aligned} \tag{23}$$

which holds since expectation operator $\mathbb{E}(.)$ is a linear operator and by assuming statistical independence between $\xi_k$ and $Ey_k$. On the one hand, by using again the properties of linearity of $\mathbb{E}$ and since $E$ is seen as a constant by $\mathbb{E}(.)$, one can show that $\mathbb{E}\left(Ey_ky_k^T E^T\right) = EC_kE^T$. One the other hand, since $\eta_k \sim \mathcal{N}(0,1)$, then Equation equation 23 becomes:

$$C_{k+1} = EC_kE^T + Q \tag{24}$$

where $Q = \begin{bmatrix} 0_{2\times 2} & 0_{2\times 2} \\ 0_{2\times 2} & \frac{\alpha_k\sigma^2}{(1+\tau_k)^2}I_{2\times 2} \end{bmatrix}$. Let us now look at limiting behavior of Equation equation 24, that is $\lim_{k\to\infty} C_k$. Let be $C = \lim_{k\to\infty} C_k$ the covariance matrix reached in the asymptotic regime, also referred to as stationary regime. Applying the limit on both sides of Equation equation 24, $C$ then satisfies

$$C = ECE^T + Q \tag{25}$$

Hence equation 25 is a particular case of discrete Lyapunov equation. For solving such equation, the vectorization operator denoted $\vec{\cdot}$ is applied on both sides on equation 25, this amounts to solve the following linear system:

$$(I_{4^2\times 4^2} - E\otimes E)\vec{C} = \vec{Q} \tag{26}$$

where $A \otimes B = \begin{bmatrix} a_{11}B & \cdots & a_{1n}B \\ \vdots & \ddots & \vdots \\ a_{m1}B & \cdots & a_{mn}B \end{bmatrix}$ stands for the Kronecker product. The solution is given by:

$$C = \overleftarrow{(I_{4^2\times 4^2} - E\otimes E)^{-1}\vec{Q}} \tag{27}$$

where $\overleftarrow{a}$ stands for the un-vectorized operator.

Let us note that, even for the 2-dimensional case considered in this section, the dimension of matrix $C$ rapidly growth and cannot be written in plain within this paper. For the following, we will keep its symbolic expression. The stationary matrix $C$ quantifies the spreading of the limit of the sequence $\{y_k\}$, as a direct consequence of the Brownian motion effect. Now we look at the directions that maximize the scattering of the points, in other words we are looking for the eigenvectors and the associated eigenvalues of $C$. Actually, the required information for the analysis of the step size is contained within the expression of the eigenvalues $\lambda_i(C)$. The obtained eigenvalues are rationale functions w.r.t. the parameters of the schemes, while their numerator brings less interest for us (supported further), we will focus on the their denominator. We obtained the following expressions:

$$\begin{aligned} \lambda_1(C) &= \frac{N_1(\alpha,\mu,L,\gamma,\sigma)}{D_1(\alpha,\mu,L,\gamma,\sigma)}, \\ \text{s.t.}\, D_1(\alpha,\mu,L,\gamma,\sigma) &= -L^2\alpha^3\mu - L^2\alpha^2\mu - \gamma L^2\alpha^2 + 2L\alpha^3\mu^2 + 4L\alpha^2\mu^2 + \\ &\quad 4\gamma L\alpha^2\mu + 2L\alpha\mu^2 + 8\gamma L\alpha\mu + 2\gamma^2 L\alpha + 4\gamma L\mu + 4\gamma^2 L \end{aligned} \tag{28}$$

$$\begin{aligned} \lambda_2(C) &= \frac{N_2(\alpha,\mu,L,\gamma,\sigma)}{D_2(\alpha,\mu,L,\gamma,\sigma)}, \\ \text{s.t.}\, D_2(\alpha,\mu,L,\gamma,\sigma) &= \alpha^3\mu^3 + 3\alpha^2\mu^3 + 3\gamma\alpha^2\mu^2 + 2\alpha\mu^3 + \\ &\quad 8\gamma\alpha\mu^2 + 2\gamma^2\alpha\mu + 4\gamma\mu^2 + 4\gamma^2\mu \end{aligned} \tag{29}$$

$$\begin{aligned} \lambda_3(C) &= \frac{N_3(\alpha,\mu,L,\gamma,\sigma)}{D_3(\alpha,\mu,L,\gamma,\sigma)}, \\ \text{s.t.}\, D_3(\alpha,\mu,L,\gamma,\sigma) &= \alpha^3\mu^3 + 3\alpha^2\mu^3 + 3\gamma\alpha^2\mu^2 + 2\alpha\mu^3 + \\ &\quad 8\gamma\alpha\mu^2 + 2\gamma^2\alpha\mu + 4\gamma\mu^2 + 4\gamma^2\mu \end{aligned} \tag{30}$$

$$\lambda_4(C) = \frac{N_4(\alpha, \mu, L, \gamma, \sigma)}{D_4(\alpha, \mu, L, \gamma, \sigma)},$$
$$\text{s.t.} D_4(\alpha, \mu, L, \gamma, \sigma) = -L^2\alpha^3\mu - L^2\alpha^2\mu - \gamma L^2\alpha^2 + 2L\alpha^3\mu^2 + 4L\alpha^2\mu^2 +$$
$$4\gamma L\alpha^2\mu + 2L\alpha\mu^2 + 8\gamma L\alpha\mu + 2\gamma^2 L\alpha + 4\gamma L\mu + 4\gamma^2 L \tag{31}$$

One can observe that:

1. given $\alpha, L, \mu, \gamma$ positive, the denominators of eigenvalues $\lambda_2$ and $\lambda_3$ are positive as well, unlike eigenvalues $\lambda_1$ and $\lambda_4$ for which some vertical asymptotes may appear. The latter will be studied in more details further. Note that, even if some eigenvalues share the same denominator, it is not the case for the numerator. This will be illustrated later in Figures 11 and 12 to ease the analysis.

2. Interestingly, the volatility of the noise defined by the parameter $\sigma$ does not appear within the expressions of the denominators. It gives us hint that these vertical asymptotes are due to the fact that spectral radius is getting close to 1 (discussed further in Appendix A.1.3). Moreover, the parameter $\sigma$ appears only within the numerators and based on intensive numerical tests, this parameter has a pure scaling effect onto the eigenvalues $\lambda_i(C)$ when studied w.r.t. $\alpha$ without modifying the trends of the curves.

Let us now study in more details the denominator of $\lambda_1$ and $\lambda_4$ and seek for critical step size as a function of $\gamma, \mu$ and $L$ at which a vertical asymptote may appear by solving:

$$- L^2\alpha^3\mu - L^2\alpha^2\mu - \gamma L^2\alpha^2 + 2L\alpha^3\mu^2 + 4L\alpha^2\mu^2 +$$
$$4\gamma L\alpha^2\mu + 2L\alpha\mu^2 + 8\gamma L\alpha\mu + 2\gamma^2 L\alpha + 4\gamma L\mu + 4\gamma^2 L = 0 \tag{32}$$
$$\equiv \mu(2\mu - L)\alpha^3 + (\mu + \gamma)(4\mu - L)\alpha^2 + (2\mu^2 + 8\gamma\mu + 2\gamma^2)\alpha + 4\gamma(\mu + \gamma) = 0$$

This polynomial equation in $\alpha$ has three roots:

$$\alpha_1 = \frac{-\gamma - \mu}{\mu},$$
$$\alpha_2 = \frac{\mu + \gamma - \sqrt{\gamma^2 - 6\gamma\mu + \mu^2 + 4\gamma L}}{L - 2\mu}, \tag{33}$$
$$\alpha_3 = \frac{\mu + \gamma + \sqrt{\gamma^2 - 6\gamma\mu + \mu^2 + 4\gamma L}}{L - 2\mu}.$$

First, it is obvious that first root $\alpha_1$ is negative given $\gamma, \mu$ assumed nonnegative and therefore can be disregarded. Concerning $\alpha_2$ and $\alpha_3$, those are real roots as soon as:

$$\gamma^2 - 6\gamma\mu + \mu^2 + 4\gamma L \geq 0$$
$$\equiv (\gamma - \mu)^2 - 4\gamma\mu + 4\gamma L \geq 0 \tag{34}$$
$$\equiv (\gamma - \mu)^2 \geq 4\gamma(\mu - L)$$

which is always satisfied since $\gamma > 0$ and $0 < \mu < L$ by hypothesis.

Further, it is obvious that the study must include three scenarios:

1. scenario 1: $L - 2\mu < 0$, or equivalently $\mu > L/2$. Given $\mu$ and $\gamma$ positive by hypothesis, it implies that $\alpha_3$ is negative and hence can be disregarded. It remains to check if $\alpha_2$ can be positive, it amounts to verify if

$$\mu + \gamma - \sqrt{\gamma^2 - 6\gamma\mu + \mu^2 + 4\gamma L} < 0$$
$$\equiv (\mu + \gamma)^2 < \gamma^2 - 6\gamma\mu + \mu^2 + 4\gamma L$$
$$\equiv \mu < \frac{L}{2}$$

which never holds by hypothesis. Therefore, for first scenario, there is no positive critical step size at which a vertical asymptote for the eigenvalues may appear.

2. scenario 2: $L - 2\mu > 0$, or equivalently $\mu < L/2$. Obviously, $\alpha_3$ is positive and hence shall be considered for the analysis of maximum step size for our NAG-GS method. It remains to check if $\alpha_2$ is positive, that is to verify if the numerator can be negative. We have seen at the first scenario that $\alpha_2$ is negative as soon as $\mu < \frac{L}{2}$ which is verified by hypothesis. Therefore, only $\alpha_3$ is positive.

3. scenario 3: $L - 2\mu = 0$. For such situation, the critical step size is located at $\infty$ and can be disregarded as a potential limitation in our study.

In summary, a potential critical and limiting step size only exists in the case $\mu < L/2$, or equivalently if $\kappa(A) > 2$. In this setting, the critical step size is positive and is equal to $\alpha_{\mathrm{crit}} = \frac{\mu + \gamma + \sqrt{\gamma^2 - 6\gamma\mu + \mu^2 + 4\gamma L}}{L - 2\mu}$. Figures 9 to 10 display the evolution of the eigenvalues $\lambda_i(C)$ for $1 \leq i \leq 4$ w.r.t. to $\alpha$ for the two first scenarios, that are for $\mu > L/2$ and $\mu < L/2$. For the first scenario, the parameters $\sigma, \gamma, \mu$ and $L$ have been respectively set to $\{1, 3/2, 1, 3/2\}$. For the second scenario, $\sigma, \gamma, \mu$ and $L$ have been respectively set to $\{1, 3/2, 1, 3\}$. As expected, one can observe on Figure 9 that no vertical asymptote is present. Furthermore, one can observe $\lambda_i(C)$ seem to convergence to some limit point when $\alpha \to \infty$, numerically we report that this limit point is zero, for all the values of $\gamma$ and $\sigma$ considered.

Finally, again as expected by the results presented in this section, Figure 10 shows the presence of two vertical asymptotes for the eigenvalues $\lambda_1$ and $\lambda_4$, and none for $\lambda_2$ and $\lambda_3$. Moreover, the critical step size is approximately located at $\alpha = 6$, the theoretical formula $\alpha_{\mathrm{crit}} = \frac{\mu + \gamma + \sqrt{\gamma^2 - 6\gamma\mu + \mu^2 + 4\gamma L}}{L - 2\mu}$ predicts 6. Finally, one can observe that, after the vertical asymptotes, all the eigenvalues converge to some limit points, again numerically we report that this limit point is zero, for all the values of $\gamma$ and $\sigma$ considered.

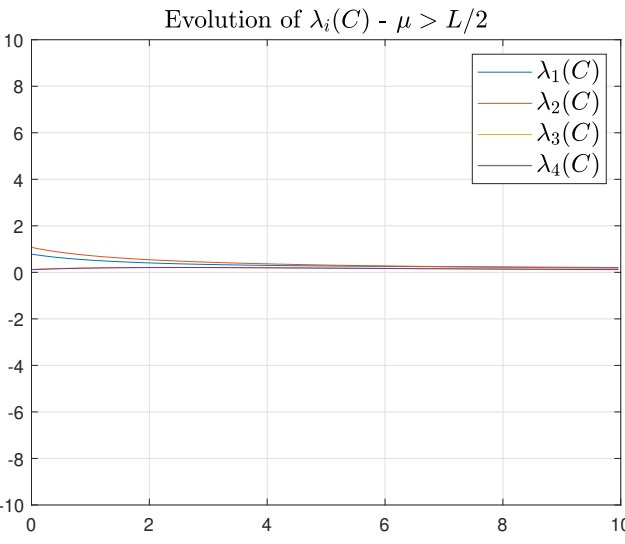

Figure 9: Evolution of $\lambda_i(C)$ w.r.t $\alpha$ for scenario $\mu > L/2$; $\sigma = 1, \gamma = 3/2, \mu = 1, L = 3/2$.

### A.1.3 A CONCLUSION FOR THE 2-DIMENSIONAL CASE

In Appendix A.1.1 and Appendix A.1.2, several theoretical results have been derived for coming up with appropriate choices of constant step size for Algorithm 1. Key insights and interesting values for the step size have been discussed from the study of the spectral radius of iteration matrix $E$ and through the analysis of the covariance matrix in the asymptotic regime. Let us summarize the theoretical results obtained:

- from the spectral radius analysis of iteration matrix $E$; two scenarios have been highlighted, that are:

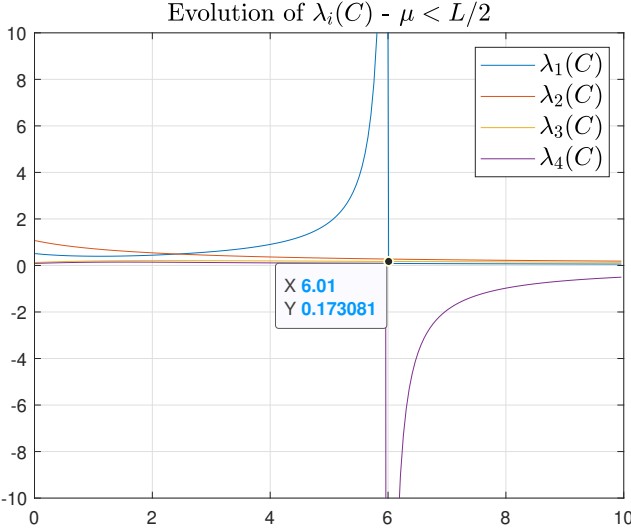

Figure 10: Evolution of $\lambda_i(C)$ w.r.t $\alpha$ for scenario $\mu < L/2$; $\sigma = 1$, $\gamma = 3/2$, $\mu = 1$, $L = 3$.

1. case $\gamma = \mu$: the step size $\alpha$ that minimize the spectral radius of matrix $E$ is $\alpha = \frac{2\mu + 2\sqrt{\mu L}}{L - \mu}$,

2. case $\gamma > \mu$: the step size $\alpha$ that minimize the spectral radius of matrix $E$ is $\alpha = \frac{\mu + \gamma + \sqrt{(\mu - \gamma)^2 + 4\gamma L}}{L - \mu}$.

- from the analysis of covariance matrix $C$ at stationarity: in the case $L - 2\mu > 0$, or equivalently $\mu < L/2$, we have seen that there is vertical asymptote for two eigenvalues of $C$ at $\alpha_{\text{crit}} = \frac{\mu + \gamma + \sqrt{\gamma^2 - 6\gamma\mu + \mu^2 + 4\gamma L}}{L - 2\mu}$, leading to an intractable scattering of the limit points $\{y_k\}_{k \to \infty}$ generated by Algorithm 1. In the case $\mu > L/2$, there is no positive critical step size at which a vertical asymptote for the eigenvalues may appear.

Therefore, for quadratic functions such that $\mu > L/2$, we can safely choose either $\alpha = \frac{2\mu + 2\sqrt{\mu L}}{L - \mu}$ when $\gamma = \mu$ either $\alpha = \frac{\mu + \gamma + \sqrt{(\mu - \gamma)^2 + 4\gamma L}}{L - \mu}$ when $\gamma > \mu$ to get the minimal spectral radius for iteration $E$ and hence the highest contraction rate for the NAG-GS method.

For quadratic functions such that $\mu < L/2$, we must show that NAG-GS method is stable for both step sizes. Let us denote by $\alpha_c = \{\frac{2\mu + 2\sqrt{\mu L}}{L - \mu}, \frac{\mu + \gamma + \sqrt{(\mu - \gamma)^2 + 4\gamma L}}{L - \mu}\}$, two values of step size for the two scenarios $\gamma = \mu$ and $\gamma > \mu$. In other to prove stability, we must show that $\rho(E(\alpha_c)) < 1$. Let us start by computing $\alpha$ such that $\rho(E(\alpha)) = 1$. As proved in Appendix A.1.6, for $\alpha \in [\alpha_c, \infty]$, $\rho(E(\alpha)) = -\lambda_4$ with $\lambda_4$ given in equation 20, we then have to compute $\alpha$ such that:

$$-\lambda_4 = -\frac{2\gamma + \alpha\gamma + \alpha\mu - L\alpha^2 + \alpha^2\mu}{2(\gamma + \alpha\gamma + \alpha\mu + \alpha^2\mu)} +$$
$$\frac{\alpha(L^2\alpha^2 - 2L\alpha^2\mu - 2L\alpha\mu - 2\gamma L\alpha - 4\gamma L + \alpha^2\mu^2 + 2\alpha\mu^2 + 2\gamma\alpha\mu + \mu^2 + 2\gamma\mu + \gamma^2)^{1/2}}{2(\gamma + \alpha\gamma + \alpha\mu + \alpha^2\mu)} = 1$$

This leads to computing the roots of a quadratic polynomial equation in $\alpha$, the positive root is:

$$\alpha = \frac{\gamma + \mu + \sqrt{4L\gamma + \gamma^2 - 6\gamma\mu + \mu^2}}{L - 2\mu} \tag{35}$$

which not surprisingly identifies to $\alpha_{\text{crit}}$ from the covariance matrix analysis [1]. Furthermore, by recalling that $\rho(E(\alpha))$ is strictly monotonically increasing function over the interval $[\alpha_c, \infty]$, showing

---

[1]It explains why the critical $\alpha$ does not include $\sigma$, this singularity is due to the spectral radius reaching the value 1.

that $\rho(E(\alpha_c)) < 1$ is equivalent to show that $\alpha_c$ is strictly lower than $\alpha_{\text{crit}}$. The formal proof is given in Lemma 2.

**Lemma 2** *Given $\gamma > 0$, and assuming $0 < \mu < L/2$, then for $\gamma = \mu$ and $\gamma > \mu$ the following inequalities respectively hold:*

$$
\frac{\mu + \gamma + \sqrt{\gamma^2 - 6\gamma\mu + \mu^2 + 4\gamma L}}{L - 2\mu} > \frac{2\mu + 2\sqrt{\mu L}}{L - \mu}
$$
$$
\frac{\mu + \gamma + \sqrt{\gamma^2 - 6\gamma\mu + \mu^2 + 4\gamma L}}{L - 2\mu} > \frac{\mu + \gamma + \sqrt{(\mu - \gamma)^2 + 4\gamma L}}{L - \mu}
\tag{36}
$$

**Proof:** Let us focus on the case $\gamma > \mu$, since $0 < \mu < L/2$ by hypothesis, second inequality from equation 36 can be written as:

$$
(L - \mu)(\gamma + \mu + \sqrt{(\gamma - \mu)^2 + 4\gamma(L - \mu)}) - (L - 2\mu)(\gamma + \mu + \sqrt{(\gamma - \mu)^2 + 4\gamma L}) > 0
$$
$$
\equiv \gamma\mu + \mu^2 + (L - \mu)\sqrt{\gamma^2 + \mu^2 + \gamma(4L - 6\mu)} + (2\mu - L)\sqrt{(\gamma - \mu)^2 + 4\gamma L} > 0
$$

Given $\gamma, \mu > 0$, it remains to show that:

$$
(L - \mu)\sqrt{\gamma^2 + \mu^2 + \gamma(4L - 6\mu)} + (2\mu - L)\sqrt{(\gamma - \mu)^2 + 4\gamma L} > 0
\tag{37}
$$

In order to show this, we study the conditions for $\gamma$ such that the left-hand side of equation 37 is positive. With simple manipulations, one can show that canceling the left-hand side of equation 37 boils down to canceling the following quadratic polynomial:

$$
(L - \mu)\sqrt{\gamma^2 + \mu^2 + \gamma(4L - 6\mu)} + (2\mu - L)\sqrt{(\gamma - \mu)^2 + 4\gamma L} = 0
$$
$$
\equiv (-3\mu + 2L)\gamma^2 + (2\mu^2 - 8L\mu + 4L^2)\gamma + 2L\mu^2 - 3\mu^3 = 0
$$

The two roots are:

$$
\gamma_1 = \frac{-\mu^2 - 2L^2 - 2\sqrt{-2\mu^4 + L^4 - 4\mu L^3 + 4\mu^2 L^2 + \mu^3 L} + 4\mu L}{2L - 3\mu}
$$
$$
\gamma_2 = \frac{-\mu^2 - 2L^2 + 2\sqrt{-2\mu^4 + L^4 - 4\mu L^3 + 4\mu^2 L^2 + \mu^3 L} + 4\mu L}{2L - 3\mu}
$$

which are reals and distinct as soon as:

$$
-2\mu^4 + L^4 - 4\mu L^3 + 4\mu^2 L^2 + \mu^3 L > 0
$$
$$
\equiv (L - 2\mu)(L - \mu)(-\mu^2 + L^2 - \mu L) > 0
$$

which holds since $0 < \mu < L/2$ by hypothesis (one can easily show that $-\mu^2 + L^2 - \mu L$ is positive in such setting). Moreover, the denominator $2L - 3\mu$ is strictly positive since $0 < \mu < L/2$. One can check that $\gamma_1$ is negative for all $\gamma, L > 0$ and $0 < \mu < L/2$ (simply show that $-\mu^2 - 2L^2 + 4\mu L$ is negative) and can be disregarded since $\gamma$ is positive by hypothesis. Therefore, proving that equation 37 holds is equivalent to show that:

$$
\gamma > \frac{-\mu^2 - 2L^2 + 2\sqrt{(L - 2\mu)(L - \mu)(-\mu^2 + L^2 - \mu L)} + 4\mu L}{2L - 3\mu}
\tag{38}
$$

It remains to show that:

$$
\mu > \frac{-\mu^2 - 2L^2 + 2\sqrt{(L - 2\mu)(L - \mu)(-\mu^2 + L^2 - \mu L)} + 4\mu L}{2L - 3\mu}
$$
$$
\equiv 0 > \mu^2 + \sqrt{(L - 2\mu)(L - \mu)(-\mu^2 + L^2 - \mu L)} - L^2 + \mu L
$$
$$
\equiv -\mu^2 + L^2 - \mu L > (L - 2\mu)(L - \mu)
$$
$$
\equiv \mu < \frac{2}{3}L
$$

which holds by hypothesis. Since $\gamma > \mu$ by hypothesis, therefore inequality equation 38 holds for any $\mu$ and $L$ positive as well, conditions satisfied by hypothesis.

Finally, since $\frac{\mu+\gamma+\sqrt{(\mu-\gamma)^2+4\gamma L}}{L-\mu} > \frac{\mu+\gamma+2\sqrt{\gamma L}}{L-\mu}$ for any $\gamma, \mu, L > 0$, then first inequality in equation 36 holds. This concludes the proof. $\qquad\square$

We conclude this section by discussing several important insights here-under:

- Except for $\alpha_{\text{crit}}$, we do not report significant information coming from the analysis of $\lambda_i(C)$ for the computation of the step size and the validity of the candidates for $\alpha$, that are $\{\frac{2\mu+2\sqrt{\mu L}}{L-\mu}, \frac{\mu+\gamma+\sqrt{(\mu-\gamma)^2+4\gamma L}}{L-\mu}\}$ respectively for the cases $\gamma = \mu$ and $\gamma > \mu$.

- Concerning the effect of the volatility $\sigma$ of the noise, we have mentioned earlier that the parameter $\sigma$ appears only within the numerators $\lambda_i(C)$ and based on intensive numerical tests, this parameter has a pure scaling effect onto the eigenvalues $\lambda_i(C)$ when studied w.r.t. $\alpha$ without modifying the trends of the curves. For compliance purpose, Figures 11 and 12 respectively show the evolution of the numerators $N_i(\alpha, \mu, L, \gamma, \sigma)$ of eigenvalues expressions of $C$ given in Equations equation 28 to equation 31 w.r.t. $\sigma$, for both scenarios $\mu < L/2$ and $\mu > L/2$. One can observe monotonic polynomial increasing behavior of $N_i(\alpha, \mu, L, \gamma, \sigma)$ w.r.t $\sigma$ for all $1 \leq i \leq 4$.

- The theoretical analysis summarized in this section is valid for the 2-dimensional case, we show in Appendix A.1.4 how to generalize our results for the $n$-dimensional case. This has no impact on the our results.

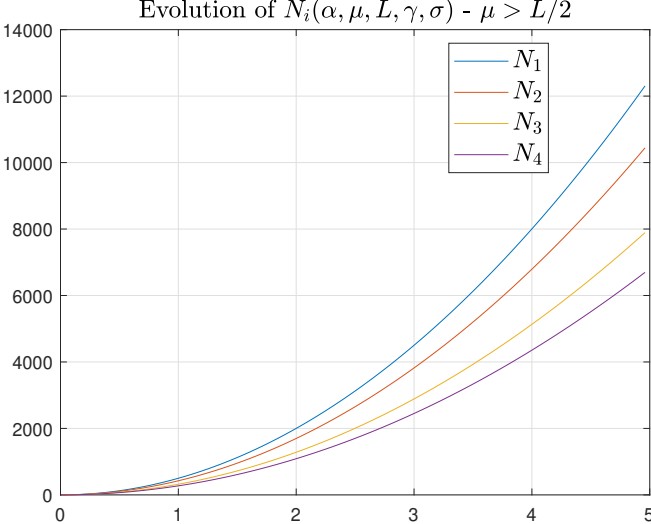

Figure 11: Evolution of $N_i(\alpha, \mu, L, \gamma, \sigma)$ w.r.t $\sigma$ for scenario $\mu > L/2$; $\gamma = 3/2$, $\mu = 1$, $L = 3/2$, $\alpha = \frac{\mu+\gamma+\sqrt{(\mu-\gamma)^2+4\gamma L}}{L-\mu}$.

### A.1.4 EXTENSION TO N-DIMENSIONAL CASE

In this section we show that we can easily extend the results gathered for the 2-dimensional case in Appendix A.1.1 and Appendix A.1.2 to the $n$-dimensional case with $n > 2$. Let us start by recalling that for NAG transformation 7, the general SDE's system to solve for the quadratic case is:

$$\dot{y}(t) = \begin{bmatrix} -I_{n\times n} & I_{n\times n} \\ 1/\gamma(\mu I_{n\times n} - A) & -\mu/\gamma I_{n\times n} \end{bmatrix} y(t) + \begin{bmatrix} 0_{n\times 1} \\ \frac{dZ}{dt} \end{bmatrix}, \quad t > 0. \tag{39}$$

Let recall that $y = (x, v)$ with $x, v \in \mathbb{R}^n$, let $n$ be even and let consider the permutation matrix $P$ associated to permutation indicator $\pi$ given here-under in two-line form:

$$\pi = \begin{bmatrix} (1 & 2) & (3 & 4) & \cdots & (n-1 & n) & (n+1 & n+2) & \cdots & (2n-1 & 2n) \\ (2*1-1 & 2*1) & (2*3-1 & 2*3) & \cdots & (2n-3 & 2n-2) & (3 & 4) & \cdots & (2n-1 & 2n) \end{bmatrix}$$

where the bottom second-half part of $\pi$ corresponds to the complementary of the bottom first half

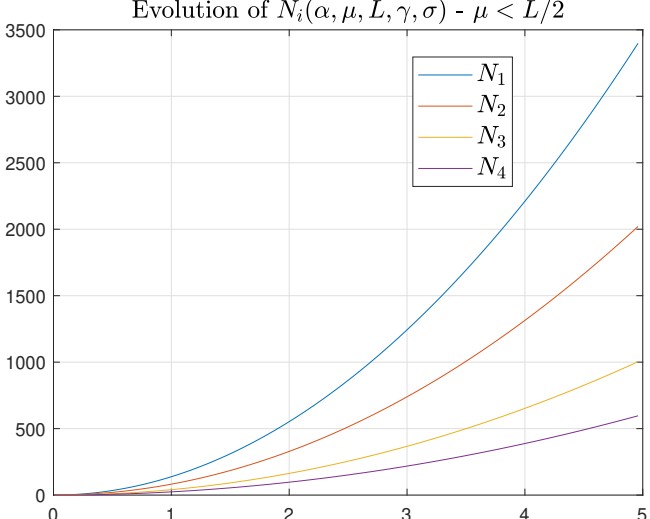

Figure 12: Evolution of $N_i(\alpha, \mu, L, \gamma, \sigma)$ w.r.t $\sigma$ for scenario $\mu < L/2$; $\gamma = 3/2$, $\mu = 1$, $L = 3$, $\alpha = \frac{\mu + \gamma + \sqrt{(\mu - \gamma)^2 + 4\gamma L}}{L - \mu}$.

w.r.t. to the set $\{1, 2, ..., 2n\}$ in the increasing order. For avoiding ambiguities, the ones element of $P$ are at indices $(\pi(1, j), \pi(2, j))$ for $1 \leq j \leq 2n$. For such convention and since permutation matrix $P$ associated to indicator $\pi$ is orthogonal matrix, equation 39 can be equivalently written as follows:

$$
\begin{aligned}
\dot{y}(t) &= PP^T \begin{bmatrix} -I_{n \times n} & I_{n \times n} \\ 1/\gamma(\mu I_{n \times n} - A) & -\mu/\gamma I_{n \times n} \end{bmatrix} PP^T y(t) + \begin{bmatrix} 0_{n \times 1} \\ \dot{Z} \end{bmatrix}, \\
&\equiv P^T \dot{y}(t) = P^T \begin{bmatrix} -I_{n \times n} & I_{n \times n} \\ 1/\gamma(\mu I_{n \times n} - A) & -\mu/\gamma I_{n \times n} \end{bmatrix} PP^T y(t) + P^T \begin{bmatrix} 0_{n \times 1} \\ \dot{Z} \end{bmatrix},
\end{aligned}
\tag{40}
$$

Since we assumed w.l.o.g. $A = \text{diag}(\lambda_1, \ldots, \lambda_n)$ with $\mu = \lambda_1 \leq \cdots \leq \lambda_j \leq \cdots \leq \lambda_n = L$, one can easily see that Equation equation 40 has the structure:

$$
\begin{bmatrix} \dot{x}_1 \\ \dot{x}_2 \\ \dot{v}_1 \\ \dot{v}_2 \\ \vdots \\ \dot{x}_{2i-1} \\ \dot{x}_{2i} \\ \dot{v}_{2i-1} \\ \dot{v}_{2i} \\ \vdots \\ \dot{x}_{n-1} \\ \dot{x}_n \\ \dot{v}_{n-1} \\ \dot{v}_n \end{bmatrix} = \begin{bmatrix} \begin{bmatrix} I_2 & -I_2 \\ 1/\gamma(\mu I_2 - A_1) & -\mu/\gamma I_2 \end{bmatrix} & 0 & 0 & 0 & 0 \\ 0 & \ddots & 0 & 0 & 0 \\ 0 & 0 & \begin{bmatrix} I_2 & -I_2 \\ 1/\gamma(\mu I_2 - A_i) & -\mu/\gamma I_2 \end{bmatrix} & 0 & 0 \\ 0 & 0 & 0 & \ddots & 0 \\ 0 & 0 & 0 & 0 & \begin{bmatrix} I_2 & -I_2 \\ 1/\gamma(\mu I_2 - A_m) & -\mu/\gamma I_2 \end{bmatrix} \end{bmatrix} \cdot \begin{bmatrix} x_1 \\ x_2 \\ v_1 \\ v_2 \\ \vdots \\ x_{2i-1} \\ x_{2i} \\ v_{2i-1} \\ v_{2i} \\ \vdots \\ x_{n-1} \\ x_n \\ v_{n-1} \\ v_n \end{bmatrix} + \begin{bmatrix} 0 \\ 0 \\ \dot{Z}_1 \\ \dot{Z}_2 \\ \vdots \\ 0 \\ 0 \\ \dot{Z}_{2i-1} \\ \dot{Z}_{2i} \\ \vdots \\ 0 \\ 0 \\ \dot{Z}_{n-1} \\ \dot{Z}_n \end{bmatrix}
\tag{41}
$$

which boils down to $m = \frac{n}{2}$ independent 2-dimensional SDE's systems where $A_i = \text{diag}(\lambda_{2i-1}, \lambda_{2i})$ with $1 \leq i \leq m$ such that $\lambda_1 = \mu$ and $\lambda_n = L$.

Therefore, the $m$ SDE's systems can be studied and theoretically solved independently with the schemes and the associated step sizes presented in previous sections. However, in practice, we will tackle the full SDE's system 39.

Let now use the "decoupled" structure given in equation 41 to come up with a general step size that will ensure the convergence of each system and hence the convergence of the full original system given in equation 39. Let us denote by $\alpha_i$ the maximum step size for the $i$-th SDE's system with $1 \leq i \leq m = n/2$. For convenience, let us consider the case $\gamma > \mu$, we apply the same method as

detailed in Appendix A.1.1 and Appendix A.1.2 to compute the expression of $\alpha_i$, we obtain:

$$\alpha_i = \frac{\mu + \gamma + \sqrt{(\mu - \gamma)^2 + 4\gamma\lambda_{2i}}}{\lambda_{2i} - \mu}$$

Finally, in Theorem 1, we show that choosing $\alpha = \frac{\mu+\gamma+\sqrt{(\mu-\gamma)^2+4\gamma L}}{L-\mu}$ ensures the convergence of NAG-GS method used to solve the SDE's system 39 in the $n$-dimensional case for $n > 2$. Theorem 1 is enunciated in Section 2.3 and the proof is given here-under.

**Proof:** Let us consider the SDE's system in the form given by equation 41 and let $\alpha_i = \frac{\mu+\gamma+\sqrt{(\mu-\gamma)^2+4\gamma\lambda_{2i}}}{\lambda_{2i}-\mu}$ be the step size selected for solving the $i$-th SDE's system with $1 \leq i \leq m = n/2$. In order to prove the convergence of the NAG-GS method by choosing a single step size $\alpha$ such that $0 < \alpha \leq \frac{\mu+\gamma+\sqrt{(\mu-\gamma)^2+4\gamma L}}{L-\mu}$, it suffices to show that:

$$\alpha = \frac{\mu + \gamma + \sqrt{(\mu - \gamma)^2 + 4\gamma L}}{L - \mu} \leq \min_{1 \leq i \leq m = n/2} \alpha_i \tag{42}$$

combined with results from Appendix A.1.3 and from Lemma 2. For proving that equation 42 holds, it sufficient to show that for any $\lambda$ such that $0 < \mu \leq \lambda \leq L < \infty$ we have:

$$\frac{\mu + \gamma + \sqrt{(\mu - \gamma)^2 + 4\gamma L}}{L - \mu} \leq \frac{\mu + \gamma + \sqrt{(\mu - \gamma)^2 + 4\gamma\lambda}}{\lambda - \mu}. \tag{43}$$

which is equivalent to show:

$$\frac{\mu + \gamma + \sqrt{(\mu - \gamma)^2 + 4\gamma L}}{L - \mu} - \frac{\mu + \gamma + \sqrt{(\mu - \gamma)^2 + 4\gamma\lambda}}{\lambda - \mu} \leq 0$$
$$\equiv \gamma\left(\frac{1}{L - \mu} - \frac{1}{\lambda - \mu}\right) + \mu\left(\frac{1}{L - \mu} - \frac{1}{\lambda - \mu}\right) + \frac{\sqrt{(\mu - \gamma)^2 + 4\gamma L}}{L - \mu} - \frac{\sqrt{(\mu - \gamma)^2 + 4\gamma\lambda}}{\lambda - \mu} \leq 0 \tag{44}$$

Since $0 < \mu \leq \lambda \leq L < \infty$ by hypothesis, one can easily show that first two terms of the last inequality are negative. It remains to show that:

$$\frac{\sqrt{(\mu - \gamma)^2 + 4\gamma L}}{L - \mu} - \frac{\sqrt{(\mu - \gamma)^2 + 4\gamma\lambda}}{\lambda - \mu} \leq 0$$
$$\equiv (-\gamma^2 - 4\gamma\lambda + 2\gamma\mu - \mu^2)L^2 + (4\gamma\lambda^2 + 2\gamma^2\mu + 2\mu^3)L + \tag{45}$$
$$\gamma^2\lambda^2 - 2\gamma^2\lambda\mu - 2\gamma\lambda^2\mu + \lambda^2\mu^2 - 2\lambda\mu^3 \leq 0$$

Note that we can easily show that the coefficient of $L^2$ is negative, hence last inequality is satisfied as soon as $L \leq \frac{-\gamma^2\lambda + 2\gamma^2\mu + 2\gamma\lambda\mu - \lambda\mu^2 + 2\mu^3}{\gamma^2 + 4\gamma\lambda - 2\gamma\mu + \mu^2}$ or $L \geq \lambda$. The latter condition is satisfied by hypothesis, this concludes the proof.

Note that one can check that $\frac{-\gamma^2\lambda + 2\gamma^2\mu + 2\gamma\lambda\mu - \lambda\mu^2 + 2\mu^3}{\gamma^2 + 4\gamma\lambda - 2\gamma\mu + \mu^2} \leq \lambda$. $\qquad\square$

The theoretical results derived in these sections along with the key insights are validated in Appendix A.1.5 through numerical experiments conducted for NAG-GS method in the quadratic case.

### A.1.5 NUMERICAL TESTS FOR QUADRATIC CASE

In this section we report some simple numerical tests for NAG-GS method (Algorithm 1) used to tackle the accelerated SDE's system given in 11 where:

- the objective function is $f(x) = (x - ce)^T A(x - ce)$ with $A \in \mathbb{S}_+^3$, $e$ a all-ones vector of dimension 3 and $c$ a positive scalar. For such strongly convex setting, since the feasible set is $V = \mathbb{R}^3$, the minimizer $\arg\min f$ uniquely exists and is simply equal to $ce$; it will be denoted further by $x^\star$. The matrix $A$ is generated as follows: $A = QAQ^{-1}$ where matrix $D$ is a diagonal matrix of size 3 and $Q$ is random orthogonal matrix. This test procedure allows us to specify the minimum and maximum eigenvalues of $A$ that are respectively $\mu$ and $L$ and hence it allows us to consider the two scenarios discussed in Appendix A.1.1, that are $\mu > L/2$ and $\mu < L/2$.

- The noise volatility $\sigma$ is set to 1, we report that this corresponds to a significant level of noise.

- Initial parameter $\gamma_0$ is set to $\mu$.

- Different values for the step size $\alpha$ will be considered in order to empirically demonstrate the optimal choice $\alpha_c$ in terms of contraction rate, but also validate the critical values for step size in the case $\mu < L/2$ and, finally, highlight the effect of the step size in terms of scattering of the final iterates generated by NAG-GS around the minimizer of $f$.

On a practical point of view, we consider $m = 200000$ points. For each of them, NAG-GS method is ran for a maximum number of iterations allowing to reach the stationarity, and the initial state $x_0$ is generated using normal Gaussian distribution. Since $f(x)$ is a quadratic function, it is expected that the points will converge to some Gaussian distribution around the minimizer $x^\star = ce$. Furthermore, since the initial distribution is also Gaussian, then it is expected than the intermediate distributions (at each iteration of NAG-GS method) are Gaussian as well. Therefore, in order to quantify the rate of convergence of NAG-GS method for different values of step size, we will monitor $\|\bar{x}^k - x^\star\|$, that is the distance between the empirical mean of the distribution at iteration $k$ and the minimizer $x^\star$ of $f$.

Figures 13 and 14 respectively show the evolution of $\|\bar{x}^k - x^\star\|$ along iteration and the final distribution of points obtained by NAG-GS at stationarity for the scenario $\mu > L/2$, for the latter the points are projected onto the three planes to have a full visualization. As expected by the theory presented in Appendix A.1.3, there is no critical $\alpha$, hence one may choose arbitrary large values for step size while NAG-GS method still converges. Moreover, the choice of $\alpha = \alpha_c$ gives the highest rate of convergence. Finally, one can observe that the distribution of limit points tighten more and more around the minimizer $x^\star$ of $f$ as the chosen step increases, as expected by the analysis of Figure 9. Hence, one may choose a very large step size $\alpha$ so that the limit points converge to $x^\star$ almost surely but at a cost of a (much) slower convergence rate. Here comes the tradeoff between the convergence rate and the limit points scattering.

Finally, Figures 15 and 16 provide similar results for the scenario $\mu < L/2$. For such scenario, the theory detailed in Appendix A.1.3 and Appendix A.1.4 predicts a critical $\alpha$ from which convergence of NAG-GS is destroyed. In order to illustrate this gradually, different values of $\alpha$ have been chosen within the set $\{\alpha_c, \alpha_c/2, (\alpha_c + \alpha_{\text{crit}})/2, 0.98\alpha_{\text{crit}}\}$. First, one can observe that the choice of $\alpha = \alpha_c$ gives again the highest rate of convergence, see Figure 15. Moreover, one can clearly see that for $\alpha \to \alpha_{\text{crit}}$, the convergence starts to fail and the spreading of the limit points tends to infinity. We report that for $\alpha = \alpha_{\text{crit}}$, NAG-GS method diverges. Again, these numerical results are fully predicted by the theory derived in previous sections.

### A.1.6 Monotonicity of Spectral radius of E for NAG-GS method

Let us recall that on $[\alpha_c, \infty]$, the spectral radius $\rho(E(\alpha))$ is equal to $|\lambda_4|$, the expression of $\lambda_4$ as a function of parameters of interests for the convergence analysis of NAG-GS method was given in equation 20 and recalled here-under for convenience:

$$\lambda_4 = \frac{2\gamma + \alpha\gamma + \alpha\mu - L\alpha^2 + \alpha^2\mu}{2(\gamma + \alpha\gamma + \alpha\mu + \alpha^2\mu)} -$$
$$\frac{\alpha\sqrt{L^2\alpha^2 - 2L\alpha^2\mu - 2L\alpha\mu - 2\gamma L\alpha - 4\gamma L + \alpha^2\mu^2 + 2\alpha\mu^2 + 2\gamma\alpha\mu + \mu^2 + 2\gamma\mu + \gamma^2}}{2(\gamma + \alpha\gamma + \alpha\mu + \alpha^2\mu)}$$
$$(46)$$

Let start by showing that $\lambda_4$ is negative on $[\alpha_c, \infty]$. Firstly, one can easily observe that the denominator of $\lambda_4$ is positive, secondly let us compute the values for $\alpha$ such that:

$$2\gamma + \alpha\gamma + \alpha\mu - L\alpha^2 + \alpha^2\mu -$$
$$\alpha\sqrt{L^2\alpha^2 - 2L\alpha^2\mu - 2L\alpha\mu - 2\gamma L\alpha - 4\gamma L + \alpha^2\mu^2 + 2\alpha\mu^2 + 2\gamma\alpha\mu + \mu^2 + 2\gamma\mu + \gamma^2} = 0$$
$$\equiv -4\gamma^2 - 4\alpha\gamma(\mu + \gamma) + \alpha^2(\gamma^2 - 4\gamma L + 2\gamma\mu + \mu^2) - \alpha^2(\gamma^2 - 4\gamma L + 6\gamma\mu + \mu^2) = 0$$
$$\equiv (-4\gamma\mu)\alpha^2 - 4\gamma(\mu + \gamma)\alpha - 4\gamma^2 = 0$$
$$(47)$$

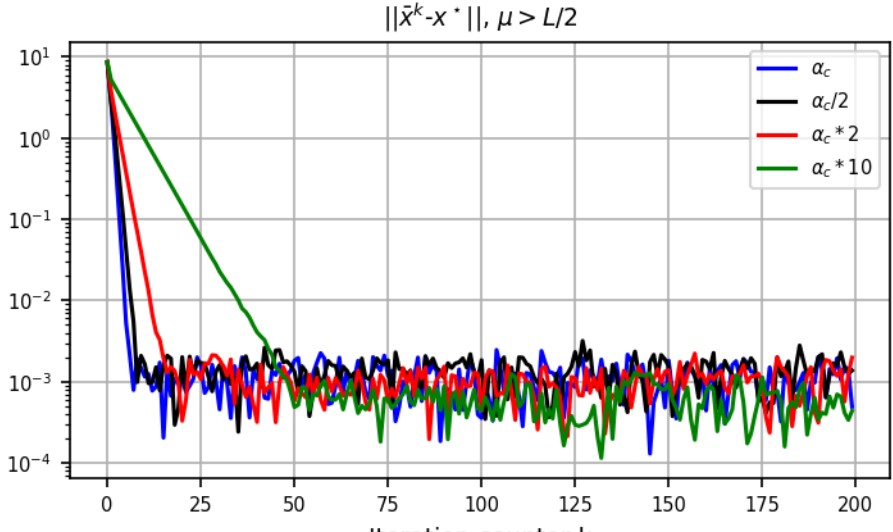

Figure 13: Evolution of $\|\bar{x}^k - x^\star\|$ along iteration for the scenario $\mu > L/2$; $c = 5$, $\gamma = \mu = 1$, $L = 1.9$ and $\sigma = 1$ for $\alpha \in \{\alpha_c, \alpha_c/2, 2\alpha_c, 10\alpha_c\}$ with $\alpha_c = \frac{2\mu + 2\sqrt{\mu L}}{L - \mu} = 5.29$.

The expression above is negative as soon as $\alpha < -1$ or $\alpha > \frac{-\gamma}{\mu} < 0$ since $\gamma, \mu > 0$ by hypothesis. The latter is always satisfied since $\alpha \geq \alpha_c > 0$ by hypothesis. Therefore $\rho(E(\alpha)) = -\lambda_4$ for $\alpha \in [\alpha_c, \infty]$.

To show the monotonic increasing behavior of $\rho(E(\alpha))$ w.r.t. $\alpha \in [\alpha_c, \infty]$, it remains to show that:

$$\frac{d(\rho(E(\alpha)))}{d\alpha} = \frac{d(-\lambda_4)}{d\alpha} > 0. \tag{48}$$

To ease the analysis, let us decompose $-\lambda_4(\alpha) = t_1(\alpha) + t_2(\alpha)$ such that:

$$t_1(\alpha) = -\frac{2\gamma + \alpha\gamma + \alpha\mu - L\alpha^2 + \alpha^2\mu}{2(\gamma + \alpha\gamma + \alpha\mu + \alpha^2\mu)}$$

$$t_2(\alpha) = \frac{\alpha\sqrt{L^2\alpha^2 - 2L\alpha^2\mu - 2L\alpha\mu - 2\gamma L\alpha - 4\gamma L + \alpha^2\mu^2 + 2\alpha\mu^2 + 2\gamma\alpha\mu + \mu^2 + 2\gamma\mu + \gamma^2}}{2(\gamma + \alpha\gamma + \alpha\mu + \alpha^2\mu)} \tag{49}$$

Let us now show that $\frac{dt_1(\alpha)}{d\alpha} > 0$ and $\frac{dt_2(\alpha)}{d\alpha} > 0$ for any $L > \mu > 0$. We first obtain:

$$\begin{aligned}\frac{dt_1(\alpha)}{d\alpha} &= \frac{(2\gamma + 2\mu + 4\alpha\mu)(2\gamma + \alpha\gamma + \alpha\mu - L\alpha^2 + \alpha^2\mu)}{(2\gamma + 2\alpha\gamma + 2\alpha\mu + 2\alpha^2\mu)^2} - \\ &\quad \frac{\gamma + \mu - 2L\alpha + 2\alpha\mu}{2\gamma + 2\alpha\gamma + 2\alpha\mu + 2\alpha^2\mu} \\ &= \frac{(L\alpha^2 + \gamma)(\gamma + \mu) + 2\alpha\gamma(L + \mu)}{2(\alpha + 1)^2(\gamma + \alpha\mu)^2}\end{aligned} \tag{50}$$

which is strictly positive since $L > \mu > 0$ and $\gamma > 0$ by hypothesis. Furthermore:

$$\frac{dt_2(\alpha)}{d\alpha} =$$
$$\frac{(\gamma + \mu)(L - \mu)(\alpha^3 L - 3\alpha\gamma) + \alpha^2(L(-\gamma^2 - \mu^2) + 2\gamma(L^2 - L\mu + \mu^2)) + \gamma(\gamma^2 - 2\gamma(2L - \mu) + \mu^2)}{2(\alpha + 1)^2(\alpha\mu + \gamma)^2\sqrt{\alpha^2(L^2 - 2L\mu + \mu^2) - 2\alpha(\gamma + \mu)(L - \mu) + \gamma^2 - 2\gamma(2L - \mu) + \mu^2}} \tag{51}$$

The remaining of the demonstration is significantly long and technically heavy in the case $\gamma > \mu$. Then we limit the last part of the demonstration for the case $\mu = \gamma$ for which we have shown previously than $\alpha_c = \frac{\mu + \gamma + 2\sqrt{\gamma L}}{L - \mu} = \frac{2\mu + 2\sqrt{\mu L}}{L - \mu}$. In practice, with respect to NAG-GS method

summarized by Algorithm 1, $\gamma$ quickly decrease to $\mu$ and equality $\mu = \gamma$ holds for most part of the iterations of the Algorithm, hence this case is more important to detail here.

The first term of the numerator of Equation 51 is positive as soon as $\alpha \geq \sqrt{\frac{3\gamma}{L}}$. In the case $\mu = \gamma$, we determine the conditions under which the second term of the the numerator of Equation 51 is positive, that is:

$$
\begin{aligned}
&\alpha^2(L(-2\mu^2) + 2\mu(L^2 - L\mu + \mu^2)) + \mu(2\mu^2 - 2\mu(2L - \mu)) > 0 \\
&\equiv \alpha^2(L(-2\mu^2) + 2\mu(L^2 - L\mu + \mu^2)) > \mu(-2\mu^2 + 2\mu(2L - \mu))
\end{aligned}
\tag{52}
$$

First one can see that:

$$
\begin{aligned}
(L(-2\mu^2) + 2\mu(L^2 - L\mu + \mu^2)) &> 0, \\
\mu(-2\mu^2 + 2\mu(2L - \mu)) &> 0
\end{aligned}
\tag{53}
$$

hold as soon as $L > \mu > 0$ which is satisfied by hypothesis. Therefore, the second term of the numerator of Equation 51 is positive as soon as

$$
\alpha > \sqrt{\frac{\mu(-2\mu^2 + 2\mu(2L - \mu))}{(L(-2\mu^2) + 2\mu(L^2 - L\mu + \mu^2))}} = \sqrt{\frac{2\mu}{L - \mu}}
\tag{54}
$$

which exists since $L > \mu > 0$ by hypothesis (the second root of equation 53 being negative). Finally, since $\alpha \in [\alpha_c, \infty]$ by hypothesis, $\frac{dt_2(\alpha)}{d\alpha}$ is positive as soon as:

$$
\begin{aligned}
\alpha_c &> \sqrt{\frac{3\mu}{L}} \\
\alpha_c &> \sqrt{\frac{2\mu}{L - \mu}}
\end{aligned}
\tag{55}
$$

hold with $\alpha_c = \frac{2\mu + 2\sqrt{\mu L}}{L - \mu}$. One can easily show that both inequalities hold as soon as $L > \mu > 0$ which is satisfied by hypothesis. This concludes the proof of the strict monotonicity of $\rho(E(\alpha))$ w.r.t. $\alpha$ for $\alpha \in [\alpha_c, \infty]$ assuming $L > \mu > 0$ and $\gamma = \mu$.

### A.2 FULLY-IMPLICIT SCHEME

In this section we present an iterative method based on the NAG transformation $G_{NAG}$ 7 along with a fully implicit discretization to tackle equation 4 in the stochastic setting, the resulting method shall be referred to as "NAG-FI" method. We propose the following discretization for equation 6 perturbated with noise; given step size $\alpha_k > 0$:

$$
\begin{aligned}
\frac{x_{k+1} - x_k}{\alpha_k} &= v_{k+1} - x_{k+1}, \\
\frac{v_{k+1} - v_k}{\alpha_k} &= \frac{\mu}{\gamma_k}(x_{k+1} - v_{k+1}) - \frac{1}{\gamma_k}Ax_{k+1} + \sigma\frac{W_{k+1} - W_k}{\alpha_k}.
\end{aligned}
\tag{56}
$$

As done for the NAG-GS method, on a practical point of view, we will use $W_{k+1} - W_k = \Delta W_k = \sqrt{\alpha_k}\eta_k$ where $\eta_k \sim \mathcal{N}(0, 1)$, by the properties of the Brownian motion.

In the quadratic case, that is $f(x) = \frac{1}{2}x^T A x$, solving equation 56 is equivalent to solve:

$$
\begin{bmatrix} x_k \\ v_k + \sigma\sqrt{\alpha_k}\eta_k \end{bmatrix} = \begin{bmatrix} (1 + \alpha_k)I & -\alpha_k I \\ \frac{\alpha_k}{\gamma_k}(A - \mu I) & (1 + \frac{\alpha_k\mu}{\gamma_k})I \end{bmatrix} \begin{bmatrix} x_{k+1} \\ v_{k+1} \end{bmatrix}
\tag{57}
$$

where $\eta_k \sim \mathcal{N}(0, 1)$. Furthermore, the parameter equation 8 is again discretized implicitly:

$$
\frac{\gamma_{k+1} - \gamma_k}{\alpha_k} = \mu - \gamma_{k+1}, \quad \gamma_0 > 0.
\tag{58}
$$

As done for NAG-GS method, heuristically, for general $f \in \mathcal{S}_{L,\mu}^{1,1}$ with $\mu \geq 0$, we just replace $Ax_{k+1}$ in equation 56 with $\nabla f(x_{k+1})$ and obtain the following NAG-FI scheme:

$$
\begin{aligned}
\frac{x_{k+1} - x_k}{\alpha_k} &= v_{k+1} - x_{k+1}, \\
\frac{v_{k+1} - v_k}{\alpha_k} &= \frac{\mu}{\gamma_k}(x_{k+1} - v_{k+1}) - \frac{1}{\gamma_k}\nabla f(x_{k+1}) + \sigma\frac{W_{k+1} - W_k}{\alpha_k}.
\end{aligned}
\tag{59}
$$

From the first equation, we get $v_{k+1} = \frac{x_{k+1}-x_k}{\alpha_k} + x_{k+1}$ that we substitute within the second equation, we obtain:

$$x_{k+1} = \frac{v_k + \tau_k x_k - \frac{\alpha_k}{\gamma_k}\nabla f(x_{k+1}) + \sigma\sqrt{\alpha_k}\eta_k}{1 + \tau_k} \tag{60}$$

with $\tau_k = 1/\alpha_k + \mu/\gamma_k$.

Computing $x_{k+1}$ is then equivalent to computing a fixed point of the operator given by the righ-hand side of equation 60. Hence, it is also equivalent to finding the root of function:

$$g(u) = u - \left( \frac{v_k + \tau_k x_k - \frac{\alpha_k}{\gamma_k}\nabla f(u) + \sigma\sqrt{\alpha_k}\eta_k}{1 + \tau_k} \right) \tag{61}$$

with $g : \mathbb{R}^n \to \mathbb{R}^n$. In order to compute the root of this function, we consider a classical Newton-Raphson procedure detailed in Algorithm 2. In Algorithm 2, $J_g(.)$ denotes the Jacobean operator

---

**Algorithm 2** Newton-Raphson method

---

**Input:** Choose the point $u_0 \in \mathbb{R}^n$, some $\alpha_k, \gamma_k, \tau_k > 0$.
    **for** $i = 0, 1, \dots$ **do**
        Compute $J_g(u_i) = I_n + \frac{\alpha_k}{\gamma_k(1+\tau_k)}\nabla^2 f(u_i)$
        Compute $g(u_i)$ using equation 61
        Set $u_{i+1} = u_i - [J_g(u_i)]^{-1}g(u_i)$
    **end for**

---

of function $g$ equation 61 w.r.t. $u$, $I_n$ denotes the identity matrix of size $n$ and $\nabla^2 f$ designates the Hessian matrix of objective function $f$. Note that the iterative method detailed in Algorithm 2 corresponds to a particular variant of second-order methods known as "Levenberg-Marquardt" Levenberg (1944); Marquardt (1963) applied to the unconstrained minimization problem $\min_{x \in \mathbb{R}^n} f(x)$ for a twice-differentiable function $f$. Finally NAG-FI method is summarized by Algorithm 3.

---

**Algorithm 3** NAG-FI Method

---

**Input:** Choose the point $x_0 \in \mathbb{R}^n$, set $v_0 = x_0$, some $\sigma \geq 0$, $\mu \geq 0$, $\gamma_0 > 0$.
    **for** $k = 0, 1, \dots$ **do**
        Sample $\eta_k \sim \mathcal{N}(0, 1)$
        Choose $\alpha_k > 0$
        Set $\gamma_{k+1} := \frac{\gamma_k + \alpha_k \mu}{1 + \alpha_k}$
        Set $\tau_{k+1} = 1/\alpha_k + \mu/\gamma_{k+1}$
        Compute the root $u$ of equation 61 by using Algorithm 2
        Set $x_{k+1} = u$
    **end for**

---

By following similar stability analysis as the one performed for NAG-GS, one can show that this method is unconditionnaly A-stable as expected by the theory of implicit schemes. In particular, one can show that eigenvalues of the iterations matrix are positive decreasing functions w.r.t. step size $\alpha$, allowing then the choice of any positive value for $\alpha$. Similarly, one can show that the eigenvalues of the covariance matrix at stationarity associated to NAG-FI method are decreasing functions w.r.t. $\alpha$ that tend to 0 as soon as $\lim \alpha \to \infty$. It implies that Algorithm 3 is theoretically able to generate iterates that converge to $\arg\min f$ almost surely, even in the stochastic setting with potentially quadratic rate of converge. This theoretical result is quickly highlighted in Figure 17 that shows the final distribution of points generated by NAG-FI once used in test setup detailed in Appendix A.1.5, in the most interesting and critical scenario $\mu < L/2$. As expected we report $\alpha$ can be chosen as big as desired, we choose here $\alpha = 1000\alpha_c$. Moreover, for increasing value of $\alpha$, the final distributions of points are more and more concentrated on the $\arg\min f$.

Therefore, NAG-FI method constitutes a good basis for deriving efficient second-order methods for tackling stochastic optimization problems, which is hard to find in the current SOTA. Indeed, second-order methods and more generally some variants of preconditioned gradient methods have recently been proposed and used in the deep learning community for the training of NN for instance.

However, it appears that there is limited empirical success for such methods when used for training NN when compared to well-tuned Stochastic Gradient Descent schemes, see for instance Botev et al. (2017); Zeiler (2012). To the best of our knowledge, no theoretical explanations have been brought to formally support these empirical observations. This will be part of our future research directions.

Besides these nice preliminar theoretical results and numerical observations for small dimension problems, there is a limitation of NAG-FI method that comes from the numerical feasibility for computing the root of the non-linear equation 61 that can be very challenging in practice. We will try to address this issue in future works.

## B  CONVERGENCE TO THE STATIONARY DISTRIBUTION

Another way to study convergence of the proposed algorithms is to consider Fokker - Planck equation for the density function $\rho(t, x)$. We will consider the simple case of the scalar SDE for the stochastic gradient flow (similarly as in equation 11). Here $f : \mathbb{R} \to \mathbb{R}$:

$$dx = -\nabla f(x)dt + dZ = -\nabla f(x)dt + \sigma dW, \quad x(0) \sim \rho(0, x).$$

It is well known, that the density function for $x(t) \sim \rho(t, x)$ satisfies the corresponding Fokker - Planck equation:

$$\frac{\partial \rho(t, x)}{\partial t} = \nabla \left( \rho(t, x) \nabla f(x) \right) + \frac{\sigma^2}{2} \Delta \rho(t, x) \tag{62}$$

For the equation 62 one could write down the stationary (with $t \to \infty$) distribution

$$\rho^*(x) = \lim_{t \to \infty} \rho(t, x) = \frac{1}{Z} \exp\left( -\frac{2}{\sigma^2} f(x) \right), \quad Z = \int_{x \in V} \exp\left( -\frac{2}{\sigma^2} f(x) \right) dx. \tag{63}$$

It is useful to compare different optimization algorithms in terms of convergence in the probability space, because it allows to study the methods in the non-convex setting. We have to address two problems with this approach. Firstly, we need to specify some distance functional between current distribution $\rho_t = \rho(t, x)$ and stationary distribution $\rho^* = \rho^*(x)$. Secondly, we don't need to have access to the densities $\rho_t, \rho^*$ themselves.

For the first problem we will consider the following distance functionals between probability distributions in the scalar case:

- **Kullback - Leibler divergence.** Several studies dedicated to convergence in probability space are available Arnold et al. (2001); Chewi et al. (2020); Lambert et al. (2022). We used approach proposed in Pérez-Cruz (2008) to estimate KL divergence between continuous distributions based on their samples.

- **Wasserstein distance.** Wasserstein distance is relatively easy to compute for scalar densities. Also, it was shown, that stochastic gradient process with constant learning rate is exponentially ergodic in the Wasserstein senseLatz (2021).

- **Kolmogorov - Smirnov statistics.** We used the two-sample Kolmogorov-Smirnov test for goodness of fit.

To the best of our knowledge, the explicit formula for the stationary distribution of Fokker-Planck equations for the ASG SDE equation 11 remains unknown. That is why we've decided to get samples from the empirical stationary distributions using Euler-Maruyama integration Maruyama (1955) with a small enough stepsize of corresponding SDE with a bunch of different independent initializations.

We tested two functions, which are presented on the figure 18. We initially generated 100

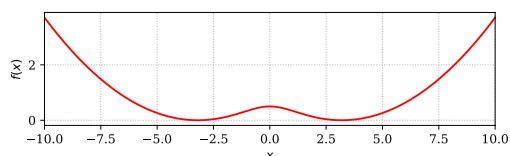

(a) Two pits function.

$$f_1(x) = \frac{1}{50} \left( 2\log\left(\cosh(x)\right) - 5 \right)^2$$

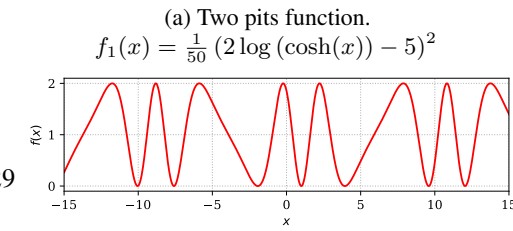

(b) Frequently modulated sin function

points uniformly in the function domain. Then we independently solved initial value problem in equation 11 for each of them with Maruyama (1955). Results of the integration are presented on the figure 19. One can see, that in relatively easy scenario 18a, NAG-GS converges faster 19a, than gradient flow to its stationary distribution, while in hard scenario 18b, NAG-GS is more robust to the large stepsize. 19b.

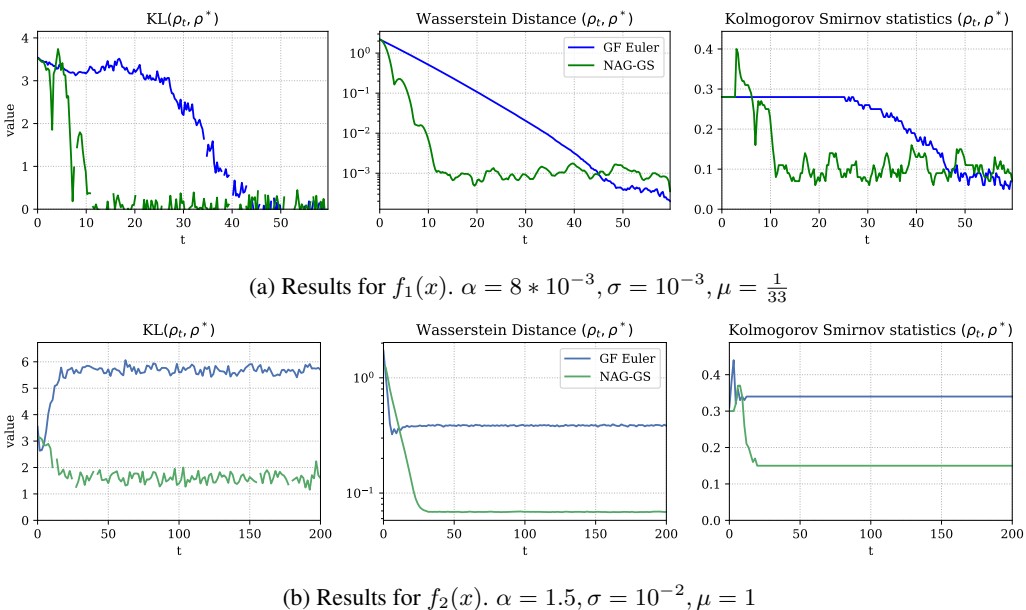

(a) Results for $f_1(x)$. $\alpha = 8 * 10^{-3}, \sigma = 10^{-3}, \mu = \frac{1}{33}$

(b) Results for $f_2(x)$. $\alpha = 1.5, \sigma = 10^{-2}, \mu = 1$

Figure 19: Convergence in probabilities of Euler integration of Gradient Flow (GF Euler) and NAG-GS for the non-convex scalar problems.

## C   ADDITIONAL EXPERIMENTAL DETAILS

In this section we provide additional experimental details. In particular, we discuss a little bit more our experimental setup and give some insights about NAG-GS as well.

Our computational resources are limited to a single Nvidia DGX-1 with 8 GPUs Nvidia V100. Almost all experiments were carried out on a single GPU. The only exception is for the training of ResNet50 on ImageNet which used all 8 GPUs.

### C.1   FINE-TUNING ROBERTA ON GLUE

After completion of experiments on toy models, we applied NAG-GS for the training of large neural networks that could be used in practice. We performed a grid search to come up with an optimal learning rate $\alpha$ for RoBERTa model (see Table 4). It is worth mentioning that there is a stability region for NAG-GS which is visible to the naked eye, we highlighted in blue some cells from Table 4 to support this claim. Furthermore, based on these simple experiments, it appears that choosing $\alpha \sim 10^{-1}$ is a good way to go for a wide variety of tasks.

In the subsequent experiments we limited ourselves to RoBERTa's training on small GLUE tasks in order to study relations between $\alpha$ and $\gamma$ in the context of high-dimensional non-convex objective functions (see Figure 20). One can see that there is oblong area in which NAG-GS converges pretty well. The ridge of this area can be empirically described with linear equation $\log \alpha = a \log \gamma + b$ in logarithmic domain. In Appendix C.2, we perform similar study for other types of model.

Table 4: Performance in fine-tuning on GLUE benchmark for different learning rate $\alpha$, fixed moment $\mu = 1$, and factor $\gamma \in \{1, 1.5\}$. Largest metric values for NAG-GS optimizer is highlighted with bold typeface. Performance metric is accuracy for all tasks with exception Matthews correlation for CoLA and Pearson correlation for STS-B. Dash in cell means an absence of a run. Higher is better.

| $\gamma$ | $\alpha$ | CoLA | MNLI | MRPC | QNLI | QQP | RTE | SST2 | STS-B | WNLI |
|---|---|---|---|---|---|---|---|---|---|---|
| 1.0 | 8E-01 | 0.00 | — | 68.38 | 50.54 | — | 52.71 | 88.42 | 19.86 | **56.34** |
| | 4E-01 | 0.00 | — | 81.13 | 50.54 | — | 52.71 | 50.92 | 87.36 | **56.34** |
| | 2E-01 | 51.16 | — | 70.59 | 88.93 | — | 52.71 | 92.55 | 89.34 | **56.34** |
| | 1E-01 | **59.64** | — | **88.73** | 90.39 | **91.09** | 52.71 | 94.27 | 89.97 | **56.34** |
| | 8E-02 | 59.41 | — | 86.27 | 91.49 | — | **72.56** | 93.92 | 90.48 | **56.34** |
| | 4E-02 | 59.16 | — | 87.75 | 91.62 | — | 72.56 | 94.38 | **90.59** | **56.34** |
| | 2E-02 | 57.13 | — | 88.48 | **91.96** | — | 67.87 | **94.72** | 90.35 | **56.34** |
| | 1E-02 | 53.21 | — | 86.52 | 91.76 | 89.36 | 55.23 | 93.92 | 88.91 | **56.34** |
| | 8E-03 | 53.94 | — | 86.27 | — | — | 54.51 | 93.58 | 89.01 | **56.34** |
| | 4E-03 | 49.25 | — | 78.92 | — | — | 53.07 | 93.69 | 86.38 | **56.34** |
| | 2E-03 | 40.73 | — | 68.38 | — | — | 52.71 | 93.35 | 79.56 | **56.34** |
| | 1E-03 | 0.00 | — | 68.38 | 86.88 | 86.01 | 52.71 | 93.35 | 58.97 | 43.66 |
| 1.5 | 8E-01 | 0.00 | 35.45 | 68.38 | 50.54 | 89.33 | 52.71 | 91.17 | 24.84 | **56.34** |
| | 4E-01 | 0.00 | 80.29 | 68.38 | 50.54 | 90.23 | 52.71 | 91.86 | 88.62 | **56.34** |
| | 2E-01 | 0.00 | 82.01 | 84.80 | 90.04 | 91.01 | 52.71 | 93.46 | 89.44 | **56.34** |
| | 1E-01 | **61.48** | 86.94 | 87.75 | 50.54 | 90.91 | 72.56 | 94.38 | 90.07 | **56.34** |
| | 8E-02 | 57.77 | 87.06 | 86.76 | 90.92 | **90.92** | **73.65** | 94.27 | **90.21** | **56.34** |
| | 4E-02 | 59.97 | **87.24** | **89.71** | **92.42** | 90.59 | 70.04 | **94.50** | 90.11 | **56.34** |
| | 2E-02 | 56.50 | 87.25 | 87.75 | 91.65 | 89.89 | 61.73 | 94.15 | 90.03 | **56.34** |
| | 1E-02 | 55.08 | 86.72 | 85.29 | 91.62 | 88.91 | 55.23 | 93.69 | 88.91 | **56.34** |
| | 8E-03 | 50.79 | 86.46 | 83.58 | 90.99 | 88.92 | 56.68 | 93.35 | 87.92 | **56.34** |
| | 4E-03 | 48.74 | 85.51 | 72.30 | 90.46 | 87.87 | 52.71 | 93.23 | 84.79 | **56.34** |
| | 2E-03 | 0.00 | 84.64 | 68.38 | 87.94 | 86.80 | 52.71 | 93.46 | 78.15 | 43.66 |
| | 1E-03 | 0.00 | 83.06 | 68.38 | 86.66 | 85.38 | 52.71 | 92.78 | 34.77 | 43.66 |

## C.2 PHASE DIAGRAMS

In Section 3.5 we mentioned that the lowest eigenvalues $\mu$ of approximated Hessian matrices evaluated during the training of ResNet20 model were negative. Furthermore, our theoretical analysis of NAG-GS in the convex case include some conditions on the optimizer parameters $\alpha$, $\gamma$, and $\mu$. In particular it is required that $\mu > 0$ and $\gamma \geq \mu$. In order to bring some insights about these remarks in the non-convex setting and inspired by Velikanov et al. (2022), we experimentally study the convergence regions of NAG-GS and sketch out the phase diagrams of convergence for different projection planes, see Figure 21.

We consider the same setup as in Section 3.3 and use hyperoptimization library OPTUNA Akiba et al. (2019). Our preliminary experiments on RoBERTa (see Appendix C.1) shows that $\alpha$ should be of magnitude $10^{-1}$. With the estimate of Hessian spectrum of ResNet20 we define the following search space

$$\alpha \sim \text{LogUniform}(10^{-2}, 10^2), \quad \gamma \sim \text{LogUniform}(10^{-2}, 10^2), \quad \mu \sim \text{Uniform}(-10, 100).$$

We sample a fixed number of triples and train ResNet20 model on CIFAR-10. Objective function is a top-1 classification error.

We report that there is a convergence almost everywhere within the projected search space onto $\alpha$-$\gamma$ plane (see Figure 21). The analysis of projections onto $\alpha$-$\mu$ and $\gamma$-$\mu$ planes brings different conclusions: there are regions of convergence for negative $\mu$ for some $\alpha < \alpha_{th}$ and $\gamma > \gamma_{th}$. Also, as it was mentioned in, there is a subdomain of negative $\mu$ comparable to a domain of positive $\mu$ in a sense of the target metrics. Moreover, the majority of sampled points are located in the vicinity of the band $\lambda_{min} < \mu < \lambda_{max}$.

Table 5: The comparision of a single step duration for different optimizers on RESNET20 on CIFAR-10. ADAM-like optimizers have in twice larger state than momentum SGDs or NAG-GS.

| OPTIMIZER | MEAN, S | VARIANCE, S | REL. MEAN | REL. VARIANCE |
|-----------|---------|-------------|-----------|---------------|
| SGD       | 0.458   | 0.008       | 1.0       | 1.0           |
| NAG-GS    | 1.648   | 0.045       | 3.6       | 5.5           |
| SGD-M     | 3.374   | 0.042       | 7.4       | 5.2           |
| SGD-MW    | 3.512   | 0.037       | 17.7      | 4.7           |
| ADAMW     | 5.208   | 0.102       | 11.4      | 12.6          |
| ADAM      | 7.919   | 0.169       | 17.3      | 20.8          |

## C.3 IMPLEMENTATION DETAILS

In our work we implemented NAG-GS in PyTorch Paszke et al. (2017) and JAX Bradbury et al. (2018); Babuschkin et al. (2020). Both implementations are used in our experiments and available online[2]. According to Algorithm 1 the size of NAG-GS state equals to number of optimization parameters which makes NAG-GS comparable to SGD with momentum. It worth to note that Adam-like optimizers has twice larger state than NAG-GS. Arithmetic complexity of NAG-GS is linear $O(n)$ in the number of parameters. Table 5 shows a comparison of computational efficiency of common optimizers used in practice. Although forward pass and gradient computations usually give the main contribution to training step, there is a settings when efficiency of gradient updates are important (e.g. batch size or a number of intermediate activations are small with respect to a number of parameters).

---

[2]https://github.com/user/nag-gs

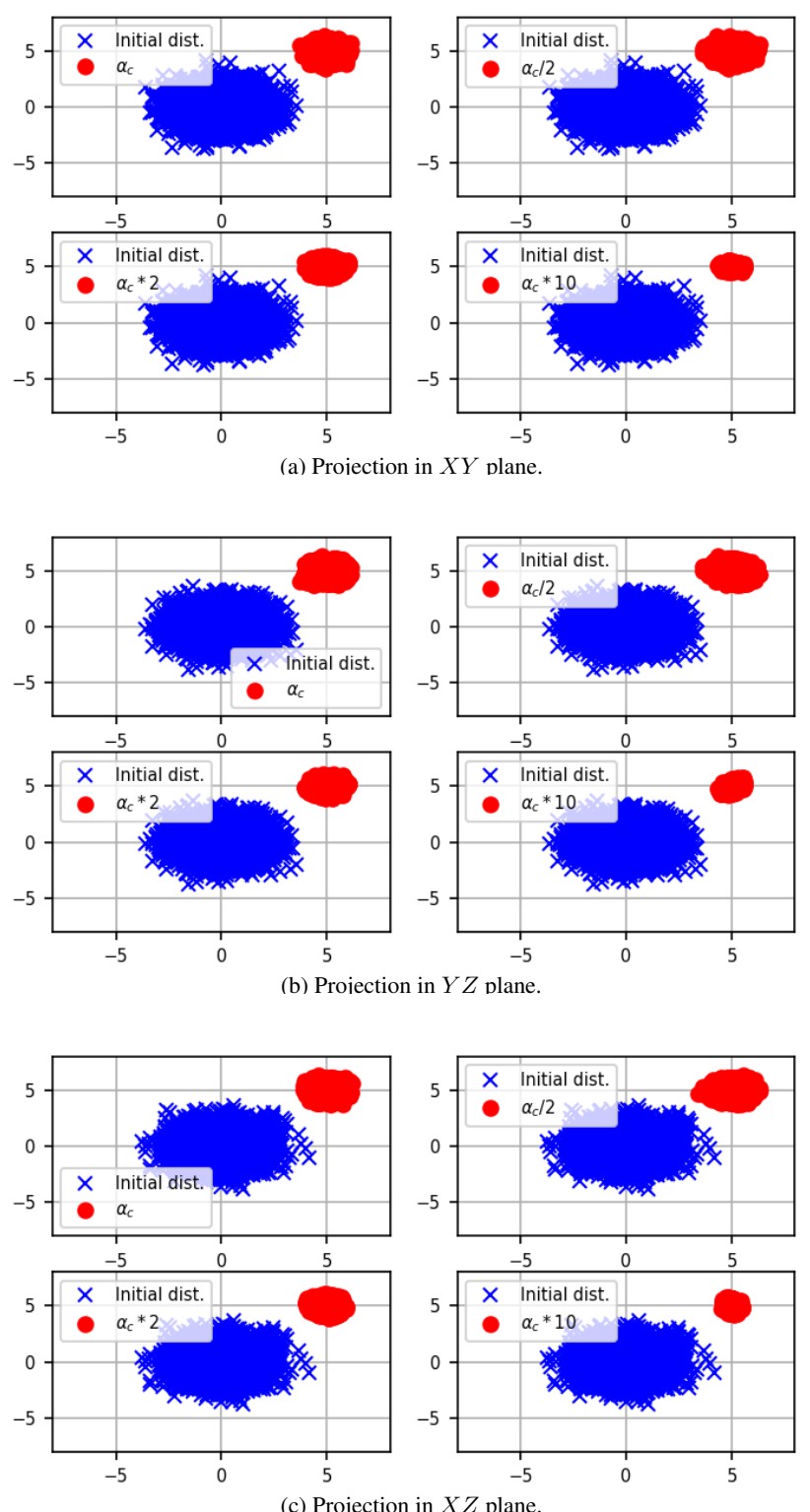

(a) Projection in $XY$ plane.

(b) Projection in $YZ$ plane.

(c) Projection in $XZ$ plane.

Figure 14: Initial (blue x) and final (red circles) distributions of points generated by NAG-GS method for the scenario $\mu > L/2$; $c = 5$, $\gamma = \mu = 1$, $L = 1.9$ and $\sigma = 1$ for $\alpha \in \{\alpha_c, \alpha_c/2, 2\alpha_c, 10\alpha_c\}$ with $\alpha_c = \frac{2\mu + 2\sqrt{\mu L}}{L - \mu} = 5.29$.

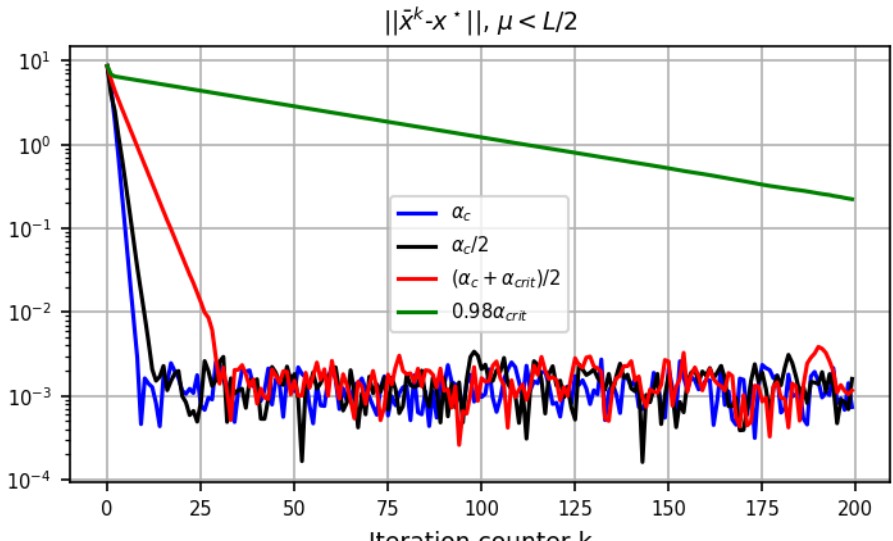

Figure 15: Evolution of $\|\bar{x}^k - x^\star\|$ along iteration for the scenario $\mu < L/2$; $c = 5$, $\gamma = \mu = 1$, $L = 3$ and $\sigma = 1$ for $\alpha \in \{\alpha_c, \alpha_c/2, (\alpha_c + \alpha_{\text{crit}})/2, 0.98\alpha_{\text{crit}}\}$ with $\alpha_c = \frac{2\mu + 2\sqrt{\mu L}}{L - \mu} = 2.73$ and $\alpha_{\text{crit}} = \frac{\mu + \gamma + \sqrt{\gamma^2 - 6\gamma\mu + \mu^2 + 4\gamma L}}{L - 2\mu} = 4.83$.

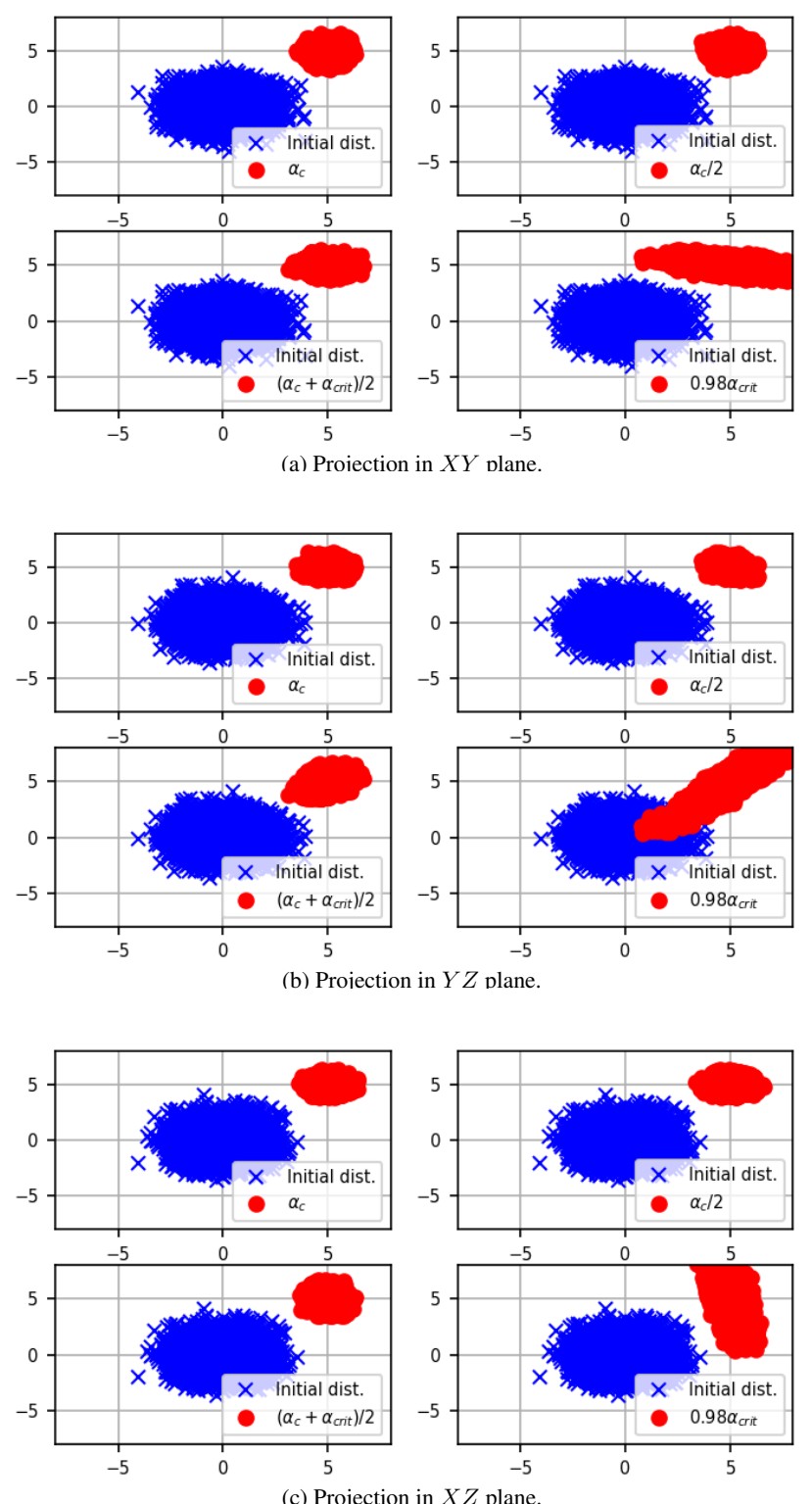

Figure 16: Initial (blue x) and final (red circles) distributions of points generated by NAG-GS method for scenario $\mu < L/2$; $c = 5$, $\gamma = \mu = 1$, $L = 3$ and $\sigma = 1$ for $\alpha \in \{\alpha_c, \alpha_c/2, (\alpha_c + \alpha_{\text{crit}})/2, 0.98\alpha_{\text{crit}}\}$ with $\alpha_c = \frac{2\mu + 2\sqrt{\mu L}}{L - \mu} = 2.73$ and $\alpha_{\text{crit}} = \frac{\mu + \gamma + \sqrt{\gamma^2 - 6\gamma\mu + \mu^2 + 4\gamma L}}{L - 2\mu} = 4.83$.

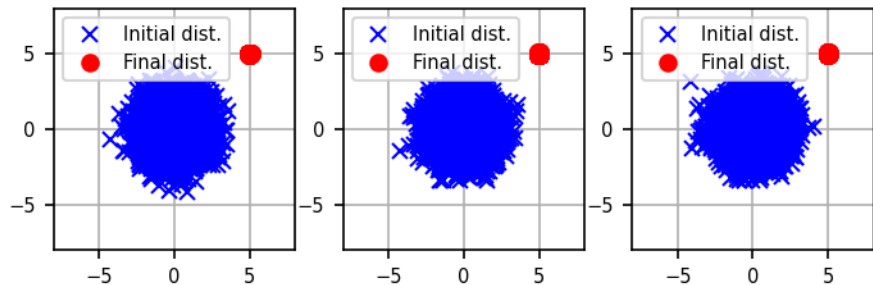

Figure 17: Projection onto $XY, YZ$ and $XZ$ planes (from left to right) of intial (blue x) and final (red circles) distributions of points gnerated by NAG-FI method - scenario $\mu < L/2$; $c = 5$, $\gamma = \mu = 1$, $L = 3$ and $\sigma = 1$ for $\alpha = 1000\alpha_c$ with $\alpha_c = \frac{2\mu + 2\sqrt{\mu L}}{L - \mu} = 2.73$.

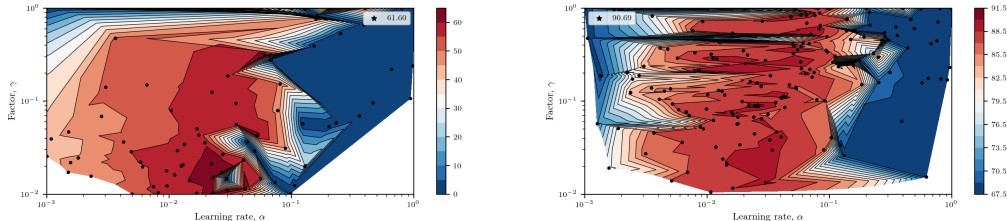

Figure 20: Landscape of accuracy metric of RoBERTa model fine-tuned on CoLA (top) MRPC (bottom) with NAG-GS. Hyperparameter optimization algorithms samples learning rate $\alpha$ from $[10^{-3}, 10^0]$ and factor $\gamma$ from $[10^{-2}, 10^0]$. Both hyperparameters are sampled from log-uniform distribution. Symbol $\star$ marks the best points.

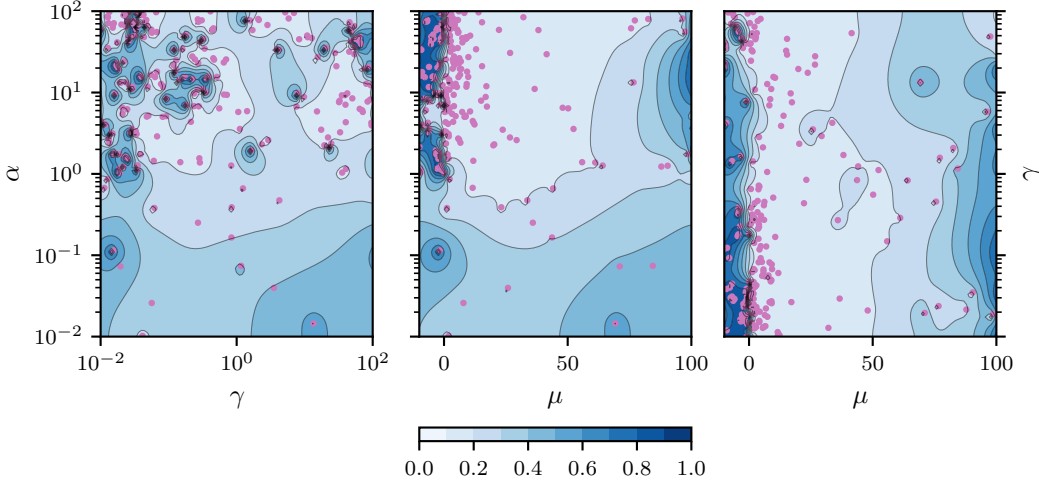

Figure 21: Landscapes of classification error for ResNet-20 model trained on CIFAR-10 with NAG-GS after projections onto $\alpha - \gamma$, $\alpha - \mu$ and $\gamma - \mu$ planes (from left to right). Hyperparameter optimization algorithm samples learning rate $\alpha$ from $[10^{-2}, 10^2]$, factor $\gamma$ from $[10^{-2}, 10^2]$, and factor $\mu$ from $[-10, 90]$. Hyperparameters $\alpha$ and $\gamma$ are sampled from log-uniform distribution, hyperparameter $\mu$ are sampled from uniform distribution.

