# OpenReview forum: "NAG-GS: semi-implicit, accelerated and robust stochastic optimizer."
_ICLR.cc/2023/Conference — Submitted to ICLR 2023_

### Official Review · Reviewer_ksyw · 2022-10-21

**Confidence:** 4
**Correctness:** 3
**Technical Novelty And Significance:** 2
**Empirical Novelty And Significance:** 2
**Recommendation:** 3

**Clarity, Quality, Novelty And Reproducibility:**

I thank the authors for good clarity. I would be able to better assess the originality and quality once the aforementioned weaknesses are discussed.

**Strength And Weaknesses:**

Strength:

(1) The Gauss-Seidel splitting and semi-implicit discretization is nice. It is great to see important knowledge in numerical analysis applied to optimization / machine learning problems.

(2) I also like the diversity of empirical tests.

Weakness:

(1) Since I could have missed the point, could the authors please explain again why the new method is important, if it produces comparable performance to existing approaches?

(2) Stochastic Gradient is not the same as (deterministic) batch-gradient plus constant covariance Gaussian noise. While there is still active research studying whether the noise of SG is Gaussian or heavy-tailed (e.g., [Gürbüzbalaban et al. The heavy-tail phenomenon in SGD. ICML 21]), at least its covariance should be state-dependent (e.g., [Li et al. Stochastic modified equations and adaptive stochastic gradient algorithms. ICML 17]). The assumption in this paper, namely an isotropic additive noise, is unrealistic. Normally I would suggest to soften the claim of “SGD”, but since an SGD version of the previous result in [Luo & Chen 2021] is the main point, I am a little confused about the contribution.

In addition, the derivation up to equation (15) is based on isotropic additive Gaussian noise, denoted by $\eta_k$, but this noise disappeared in Algorithm 1. I can’t find $\eta$ or even its amplitude $\sigma$ and became puzzled. My guess is $\nabla f$ in Algorithm 1 is no longer the deterministic full-batch gradient used before, but a stochastic gradient based on mini-batch. But if this is the case, the derivation is a little disconnected from SG (see my above point(2)).

(3) I wish the theoretical part could be stronger. The current status of the field is well beyond convergence analysis for quadratic objectives (Thm.1). In fact, if I didn’t misunderstand, Thm.1 is for a deterministic setup. Appendix B empirically investigated the stochastic setup, based on a statement that the stationary distribution of (11) remains unknown (and thus implying theoretical analysis is out of reach). But it is a standard procedure to characterize the stationary distribution: (11) is a linear SDE and its stationary distribution is just a Gaussian, whose covariance can be solved for algebraically via Ricatti equation. Besides, I missed why the part of Appendix B in page 29 discussed overdamped Langevin instead of equation (11).

Back to the convergence analysis of the full-blown nonlinear SDE, could the authors please explain why tools in, e.g., [Ma et al. Is there an analog of Nesterov acceleration for gradient-based MCMC? Bernoulli 21] and [Li et al. Hessian-free high-resolution Nesterov acceleration for sampling. ICML 22] cannot be used to quantify the speed of convergence in a more general setup?


**Summary Of The Paper:**

This paper starts with a variant of accelerated gradient flow considered in [Luo & Chen 2021], and added noise in a degenerate, underdamped Langevin alike fashion. The resulting SDE is then discretized semi-implicitly, resulting in a “stochastic gradient” optimization algorithm. “Convergence analysis” is conducted on quadratic loss. The method is then empirically tested, and performances comparable to existing optimizers were demonstrated on a collection of important downstream tasks.

**Summary Of The Review:**

The Gauss-Seidel splitting and semi-implicit discretization is definitely an interesting idea, and I strongly encourage the authors to explore it a bit more, possibly with improved theory so that the benefits could be made quantitative. For now, I’m not very sure about either its algorithmic or theoretical impact, and thus unable to provide a very positive recommendation.

---

### Official Review · Reviewer_Dog8 · 2022-10-24

**Confidence:** 3
**Correctness:** 3
**Technical Novelty And Significance:** 3
**Empirical Novelty And Significance:** 2
**Recommendation:** 5

**Clarity, Quality, Novelty And Reproducibility:**

The theory part is well presented, but not the numerical results.
The algorithm is novel, but not well supported in experiments.


**Strength And Weaknesses:**

**Strength:**
- (S1) Applying GS for discretizing SDEs is interesting and worth careful investigation.
- (S2) Theoretical analysis, though limited to quadratic problems, is still insightful to understand the behavior of NAG-GS.

**Weakness:**

The main weakness of this work lies in the numerical experiments. Confined by computational resources, the results do not support the efficiency of proposed method well. In particular, in many test cases, NAG-GS does not catch up with AdamW or SGDm. And sometimes the gap is not small. In particular,
- (W1) it can be beneficial to run tests on quadratic functions to support the theoretical findings.
- (W2) The experiment setups for logistic regression in Section 3.1 is not clear. In addition, it is more clear if the test accuracy can be summarized in a table for comparison. Moreover, when the learning rate is small, SGDm seems to have better performance over NAG-GS. For the relatively simple logistic regression problem, it does not bother too much for hyperparameter tuning. In other words, a better test accuracy is more important.
- (W3) In section 3.2, the authors write “AdamW step size is $10^{-5}$, which is much smaller than $10^{-2}$ of NAG-GS”. In practice, it is common for Adam type algorithm to use a smaller learning rate than SGD. More importantly, on more than half downstream tasks, NAG-GS fails to catch up with AdamW.
- (W4) It is unclear for the reason of switching acc@1 to acc@5 in section 3.3. Once again, NAG-GS does not offer competitive results.


**Summary Of The Paper:**

This paper proposes the NAG-GS optimizer. The algorithm is designed by considering the dynamics of an accelerated systems of SDE discretized via Gauss-Seidel (GS) splitting. NAG-GS is theoretically analyzed on quadratic problems. Numerical experiments, including logistic regression, finetuning RoBerta on downstream tasks, and image classification via ResNet, are carried out to illustrate the usefulness of the proposed method.

**Summary Of The Review:**

This paper puts more weight on the theoretical side, but the performance in typical machine learning tasks confines its practical potential.

---

### Official Review · Reviewer_AnLK · 2022-10-24

**Confidence:** 4
**Clarity, Quality, Novelty And Reproducibility:** There are a few typos, but no signifi…
**Correctness:** 4
**Technical Novelty And Significance:** 2
**Empirical Novelty And Significance:** 2
**Recommendation:** 3

**Strength And Weaknesses:**


The semi-implicit algorithm is general well-motivated and presented. However I am confused on a couple points: the equations 14 do not appear to match exactly the pseudo-code Algorithm. For example, should we not set $x_{k+1} = (1+\alpha_k)^{-1}(x_k \alpha_k v_k)$ rather than $x_{k+1} = (1-\alpha_k) x_k + \alpha_k v_k$? Similar issue may be involved with the $v_k$ update.


In regards to the theoretical convergence results, these seem to be asymptotic statements that the algorithm will eventually converge on quadratic losses with gaussian noise in the gradients. This seems rather weak given the current state of the art in acceleration under unbiased gradient noise (e.g. see https://proceedings.mlr.press/v119/joulani20a/joulani20a-supp.pdf), which establishes non-asymptotic convergence rates for general convex smooth function under arbitrary unbiased gradient noise with bounded variance without requiring knowledge of several problem parameters.

In regards to the experiment, it seems to be slightly worse than SGD on computer vision tasks, and comparable to AdamW on transformer fine-tuning tasks. All in all, this does not seem convincing that we should switch from AdamW. Is it possible perhaps to make an argument that this method requires less tuning, as was hinted at in the text?


**Summary Of The Paper:**

This paper considers a class of accelerated algorithms and provides some analysis for noisy inputs to the algorithm. It is shown that the algorithms will converge even when presented with gradients corrupted with gaussian noise on quadratic losses. Experimental results on deep learning benchmarks show competitive performance with AdamW.


**Summary Of The Review:**

The theoretical results here seem to be in a somewhat preliminary state and the empirical results do not seem strong enough to justify acceptance on their own.

---

### Official Review · Reviewer_DiaF · 2022-10-25

**Confidence:** 2
**Correctness:** 3
**Technical Novelty And Significance:** 2
**Empirical Novelty And Significance:** 3
**Recommendation:** 5

**Clarity, Quality, Novelty And Reproducibility:**

The writing of this work is clear and well-organized and the novelty looks good to me.

**Strength And Weaknesses:**

## Strength
- This paper is well-organized and relatively easy to follow. The proofs seem to be theoretically sound. The theoretical background motivates the algorithm smoothly, and the proposed method is achieving great empirical performance.

## Weaknesses
- Although simulations are implemented for various settings, the theory only holds for the quadratic functions.
- The theory only shows NAG-GS is convergent (for quadratic functions). Other properties such as the convergence rate and improvement of the range of learning rate are missing.

===========================
I have read other reviewers' responses and authors' comments. I basically agree with other reviewers. Although applying semi-implicit discretization to SDE of NAG is novel and worth more future exploration, the current theoretical results provided in this paper are preliminary. Also, the experiments cannot convince me one should switch to NAG-GS in practice in sufficiently many settings. Hence, I'll keep my score.

**Summary Of The Paper:**

 This paper presents the NAG-GS algorithm, a Gauss-Seidel type discretization of the Nesterov-like SDE. NAG-GS maintains the acceleration nature while improving the robustness to the section of learning rate. Numerical simulations are implemented to show that NAG-GS is competitive with state-of-the-art methods.

**Summary Of The Review:**

In summary, I think this is nice work with good intuition and theoretical background that motivates the final algorithm smoothly. I hope to see more detailed theoretical results such as convergence rate and the improvement of the learning rate, as these two properties are stated as the main advantage of the proposed NAG-GS algorithm. It would be fantastic if the theory could be extended to the more general problems such as optimizing functions in $\mathcal{S}_{L, \mu, \mu}^{1,1}$.

---

### Decision · Program_Chairs · 2023-01-20

**Decision:**

Reject

**Justification For Why Not Higher Score:**

As highlighted above the reviewers unanimously found the paper to below the bar for acceptance for ICLR. The reasoning were based on the following points raised in the reviews --

1. The setting is limited (quadratics) and non-asymptotic results and algorithms for more general settings are known.
2. The experimental results showed in the paper were not convincing enough for the reviewers to confidently suggest that the proposed algorithm is better than existing algorithms in a useful setting.
3. Other limiting assumptions about the setup and the noise model.

**Justification For Why Not Lower Score:**

N/A

**Metareview: Summary, Strengths And Weaknesses:**

The paper proposes a new stochastic optimization algorithm NAG-GS obtained via Gauss-Siedel discretization of a Nesterov-like stochastic differential equation. The paper provides convergence bounds for the proposed algorithm. The reviewers unanimously found the paper to be slightly below the bar for ICLR and the main points raised were --
1. The setting is limited (quadratics) and non-asymptotic results and algorithms for more general settings are known.
2. The experimental results showed in the paper were not convincing enough for the reviewers to confidently suggest that the proposed algorithm is better than existing algorithms in a useful setting.
3. Other limiting assumptions about the setup and the noise model.